# Transposable element-mediated rearrangements are prevalent in human genomes

Parithi Balachandran [1], Isha A. Walawalkar[1], Jacob I. Flores[1], Jacob N. Dayton[1], Peter A. Audano [1] & Christine R. Beck [1,2,3] ✉

Transposable elements constitute about half of human genomes, and their role in generating human variation through retrotransposition is broadly studied and appreciated. Structural variants mediated by transposons, which we call transposable element-mediated rearrangements (TEMRs), are less well studied, and the mechanisms leading to their formation as well as their broader impact on human diversity are poorly understood. Here, we identify 493 unique TEMRs across the genomes of three individuals. While homology directed repair is the dominant driver of TEMRs, our sequence-resolved TEMR resource allows us to identify complex inversion breakpoints, triplications or other high copy number polymorphisms, and additional complexities. TEMRs are enriched in genic loci and can create potentially important risk alleles such as a deletion in *TRIM65*, a known cancer biomarker and therapeutic target. These findings expand our understanding of this important class of structural variation, the mechanisms responsible for their formation, and establish them as an important driver of human diversity.

The development of a genome reference sequence over 20 years ago has driven greater knowledge of the number, effects, and variation of transposable elements (TEs) across human genomes[1]. The current haploid human reference contains over 4 million annotated TEs[2,3], and by computational estimates TEs occupy up to two-thirds of human genome[4,5]. TEs can alter the structure of genomes through transposition (de novo insertion)[6], polymorphism[7–10], transduction[11–14], and transposition-associated rearrangements[15–18]. In addition to retrotransposition, TE copies can play major roles in shaping human genomes, promoting polymorphism, and contributing to genomic instability.

Two homologous TEs can act as substrates for ectopic DNA repair resulting in structural variant (SV) formation we collectively call TE-mediated rearrangements (TEMRs). TEMRs are responsible for deletion of ~850 kbp of the human reference genome when compared to the chimpanzee draft genome[1,19,20], and were recently found to account for a loss of ~80 kbp in a Korean genome[21]. TEMRs are implicated in the expansion of segmental duplications in human genomes[22]. TEMRs are also associated with deletions of exonic regions, leading to cancer

predisposition syndromes[23,24], and Mendelian diseases[25–27]. Furthermore, complex genomic rearrangements often harbor TEs at breakpoint junctions, and these TEMRs carry similar characteristics as simple deletions[28,29]. Studying the genes and the genomic context of TEMR-associated instability can define the role these SVs play in mediating human disease and identify loci prone to this rearrangement mechanism[30,31]. Although these analyses were important to establish the extent of TEMRs in comparative genomics and disease, the prevalence of TEMRs in SV callsets, the mechanisms that cause these rearrangements, the allele frequency of disease associated variants, and the full spectrum of how TEs contribute to SV formation are still poorly understood.

The vast majority of current studies investigating TEMR mechanisms point to non-allelic homologous recombination (NAHR) generating deletions in human genomes[30,32]. Although NAHR dominates existing genome-wide TEMR studies based on reference genomes[19,20], a few cell-culture based systems have been derived to address mechanisms of TEMRs; these studies have indicated NAHR as well as non-homologous end joining (NHEJ)[33], single-strand annealing

[1]The Jackson Laboratory for Genomic Medicine, Farmington, CT, USA. [2]Department of Genetics and Genome Sciences, University of Connecticut Health Center, Farmington, CT, USA. [3]Institute for Systems Genomics, University of Connecticut, Storrs, CT, USA. ✉e-mail: christine.beck@jax.org

(SSA)[34,35], and microhomology-mediated end joining (MMEJ)[33] in the formation of these classes of SVs. Importantly, many of these studies indicate that more diverged repeats are unlikely to generate junctions in homologous regions of the two TEs, and these data are not reflected in human rearrangements[31]. Finally, a number of recent studies have detailed rearrangements with TEs at the junctions of inversions, duplications, and complex genomic rearrangements; these studies have indicated that repair by microhomology-mediated break induced replication (MMBIR) can mediate diverse classes of TEMR[28,31,36,37]. These mechanistic interrogations are impacted by the fact that most SV studies to date either lack precision or resort to hand curation of breakpoint junctions to determine potential mechanisms of formation[19,20,31]. Thus, distinguishing the mechanisms of TEMR across large numbers of events would be prohibitive, and determining the scope of TEMRs in generating human diversity has been difficult to assess. The widespread investigation of TEMRs across genomes allows an unbiased view of the diverse types of resultant SVs and the different mechanisms that drive TEMR; this type of approach has not been applied to genome wide analyses and will help interrogate how TEs lead to genomic instability and variation.

Identification of TEMRs and accurate characterization of their breakpoint junctions is challenging. Short-read sequencing (SRS) is able to identify most deletions, including TEMRs, outside of simple repeats and segmental duplications[38–40] even with its inherent limitations[41]. Long-read sequencing (LRS) has overcome many of the challenges faced by SRS in identifying SVs across genomes[38,42]. Recent advances in LRS have enabled long and contiguous de novo assembly of genomes and detection of SVs with precise junctions within repetitive regions of the genome[16,43]. Although a majority of publicly available datasets are still SRS data[44–46], LRS data is now in production for thousands of human genomes (Human Genome Structural Variation Consortium (HGSVC), Human Pangenome Reference Consortium (HPRC), Solving the Unsolved Rare Disease (Solve-RD) and AllofUs). LRS still has significant limitations including cost and input DNA quantities, therefore, we sought to develop a comprehensive TEMR identification method that using either short-read (Illumina) or long-read (PacBio CLR) sequencing data for discovery and characterization of these events. This approach is designed to scale across many samples and will enable high-throughput mechanistic inference for a growing number of sequenced genomes.

In this study, we identify 493 nonredundant TEMRs using SRS and LRS by applying our approach to three diverse genomes[38], ascertain the precise junctions of 70 randomly selected TEMRs with PCR and Sanger sequencing, and verify the accuracy of breakpoint junctions from matching phased HiFi genome assemblies[16]. By discerning the precise junctions for all 493 TEMR events we infer mechanisms involved in the formation of TE-driven deletions, duplications, and inversions. Previously, the role of TEs in generating duplications and inversions was not appreciated. Additionally, we identify TEs mediating higher order amplifications and complex SVs indicating the range of rearrangements driven by TEs. We show that *Alu* elements are a major (80.5%) contributor to TEMRs, primarily via homologous recombination mechanisms; yet the length of the homology at the junction of these *Alu* TEMRs is shorter (median of 15 bp) than what we expect from a traditional non-allelic homologous recombination event (>100 bp). We show that TEs not only affect the genome through retrotransposition but are also a substrate for widespread rearrangements creating 635 kbp of structural alterations per human genome. As TEMRs disproportionately affect genes, they are an important source to study phenotypic variation, disease, and human evolution.

## Results

### Identifying transposable element-mediated rearrangements

To call SVs genome-wide, we analyzed SRS using Manta[47], LUMPY[48], and DELLY[49] and LRS using pbsv (https://github.com/PacificBios ciences/pbsv), Sniffles[50] and SVIM[51]. We generated a consensus call-set using these individual SRS and LRS SV callers and ensemble heuristics to maximize accuracy (Methods). We implemented our pipeline on Illumina and PacBio CLR data across three well-characterized genomes representative of: (1) population admixture, Puerto Rican HG00733 (PUR); (2) low diversity, Southern Han Chinese HG00514 (CHS); and (3) high diversity, Yoruban NA19240 (YRI)[38]. Implementing a multi-caller approach with additional filters have enabled us to significantly reduce the number of false positive in our callset (Methods). Due to the repetitive nature of TEs and the technical difficulty it causes during variant calling, analyzing TEMRs without any stringent filtering could led to an unreliable analysis due to false positive variant calls (Supplementary Fig. 1). We have demonstrated with our pipeline that an ensemble approach with simple filters can result in a reliable callset outside simple repeat regions, especially with SRS data (Supplementary Fig. 1). We obtained phased HiFi assemblies for these three samples[16], and we used a new version of PAV[16] for breakpoint homology. We merged the calls from all these methods and across all three individuals into a single nonredundant high-confidence callset of 5,297 SVs containing 4,997 deletions, 239 duplications and 61 inversions, with an average of 3,111 SVs per individual (Methods).

SVs with both breakpoints in different TEs of the same element class were categorized as TEMRs (Methods). In contrast, SVs with zero or one breakpoint within a TE, or with both breakpoints within different types of TEs were classified as non-TEMR events. From our high-confidence callset of 5,297 SVs, we identified 543 nonredundant TEMRs (10.25%) across all three individuals (Fig. 1a). We identified an average of 263 TEMRs per sample (236 from PUR, 236 from CHS, and 316 from YRI) and they collectively affected an average of 795 kbp per sample. The 543 TEMRs consisted of 11 classes of TEs: *Alu* (397), LINE-1 (96), ERVL-MaLR (14), ERV1 (11), ERVL (8), L2 (6), ERVK (3), MIR (2), SVA (2), TcMar-Mariner (2), TcMar-Tigger (1), and hAT-Charlie (1) (Supplementary Table 1). Due to the prevalence of LINE-1 and *Alu*-mediated events, the difficulties in aligning ERVs and divergent transposons to consensus sequences, and the small number of TEMRs driven by non-*Alu* or LINE-1 categories precluding extensive mechanistic work, we focused on the two primary categories of TEMR in this study (493: 397 *Alu* and 96 LINE-1). Interestingly, although 90.3% (445) TEMRs were deletions, we also identified 33 duplications and 15 inversions, classes of TEMRs that were not previously surveyed in normal human genomes (Fig. 1b).

The size of TEMRs varies greatly from 93 bp to 25,425 bp with a median length of 1342 bp (Supplementary Fig. 2), and these events are longer than the non-TEMRs (median 321 bp). Polymorphic MEIs account for a large proportion of SV calls (median 317 bp), upon excluding them we still observed a significant difference in median lengths between TEMR deletions and non-TEMR deletions (1,345 bp vs 528 bp; $p < 0.01$, Welch's $t$-test). TEMR duplications were also longer than non-TEMR duplications (1,085 bp vs 262 bp) but did not reach statistical significance. Conversely, TEMR inversions were significantly smaller than non-TEMR inversions (2,454 bp vs 8,870 bp, $p < 0.001$, Welch's $t$-test), which are generally mediated by larger segmental duplications[52] (Supplementary Fig. 3).

We examined the overall genomic architecture of TEMRs and found that 91% of deletions and duplications had junctions in TEs in the same/direct orientation and all inversions had junctions in TEs in the opposite/inverted orientation (Fig. 1b). We found that *Alu* TEMRs (median length of 1,163 bp) are typically shorter than LINE-1 TEMRs[32] (median length of 4,469 bp; $p < 1e^{-5}$, Welch's $t$-test); this includes both full-length (7,663 bp; $p < 1e^{-5}$, Welch's $t$-test) and truncated LINE-1 elements (median length of 3,618 bp; $p < 1e^{-4}$, Welch's $t$-test) (Fig. 1c). In human genomes, full-length LINE-1 elements (6 kbp)[53] are almost 20-fold longer than full-length *Alu* elements (300 bp)[54] and therefore provide a longer substrate for recombination.

## Characterization of TEMR breakpoint junctions

To identify the breakpoint junctions of TEMRs with nucleotide accuracy, we randomly selected 70 TEMRs (66 deletions and 4 duplications) for PCR and Sanger sequencing (Supplementary Table 2). By manually reconstructing TEMR breakpoint junctions with Sanger sequences (Fig. 1d), we found that 55.7% (39) of TEMRs had breakpoint homology of at least 5 bp, 27.1% (19) had 1-4 bp breakpoint homology, and 17.1% (12) showed no breakpoint homology. A higher percentage and length of breakpoint homology is expected since TEMRs are mediated by homologous sequences.

We then compared the breakpoint junctions identified by all the SV callers used in this study with the corresponding manually curated junctions. Of the 58 out of 70 events with breakpoint homology, PAV calls from HiFi phased assemblies supported 57 (98.3%) events and the imprecise junction for the one remaining event was due to a nearby SNV. For the 12 TEMRs without breakpoint homology, PAV called identical breakpoints for six TEMRs, and the remaining six TEMRs had inaccurate breakpoint junctions due to the presence of indels at the junction. We observed that Manta was able to identify the indels present at the junctions for these six TEMRs and accurately call the breakpoints. Among the read-based callers, Manta had the highest breakpoint precision of 88% (51 out of 58) for TEMRs with breakpoint homology and 91% (11 out of 12) for TEMRs without breakpoint homology, although Manta failed to discover three TEMRs. Interestingly, when Manta identifies a homology or an indel at the breakpoint junction of an event, the junction was 100% precise. The breakpoint precision statistics for all SV callers used in this study can be found in (Supplementary Table 3). Using these results as a guide, we annotated

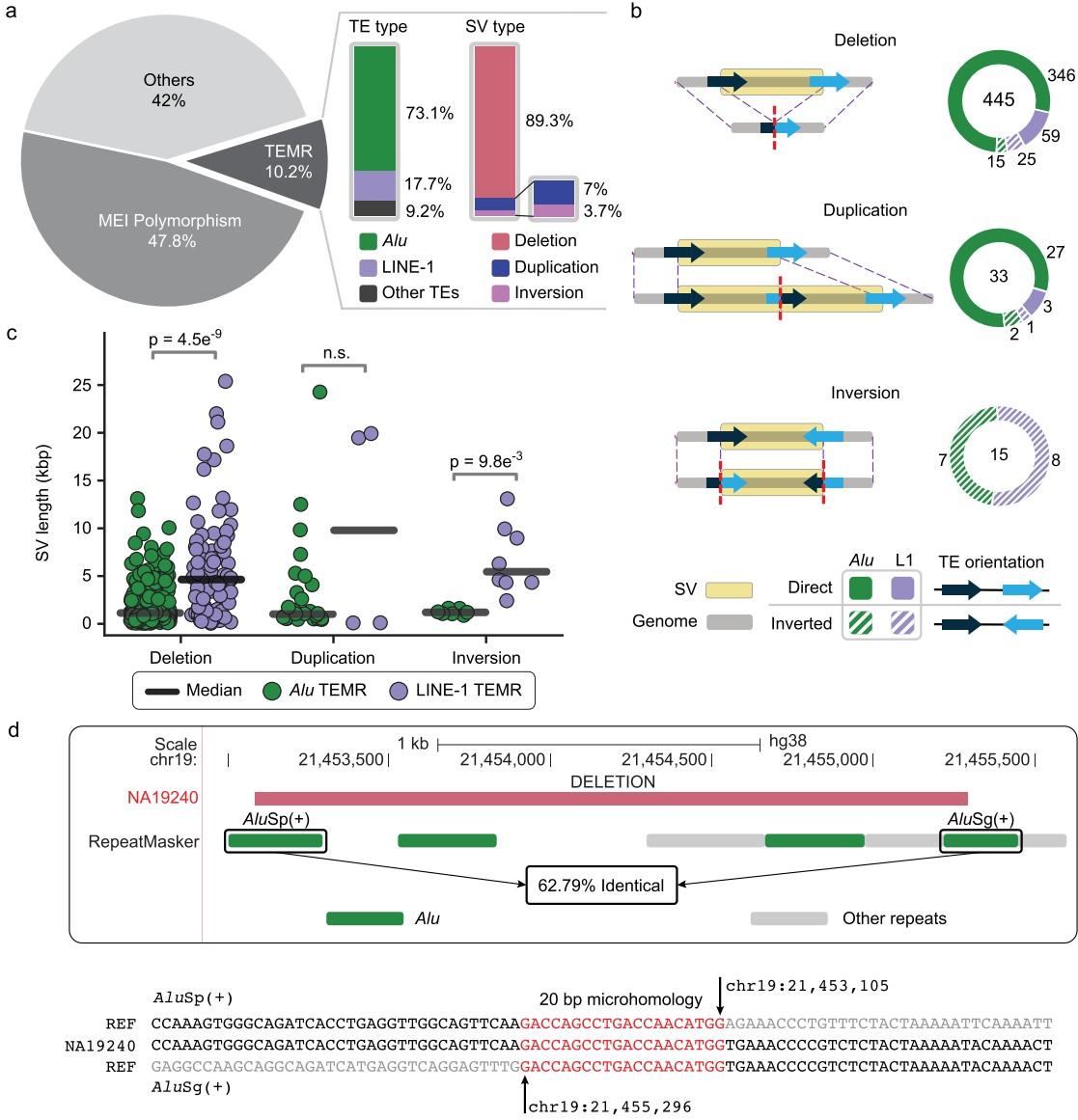

**Fig. 1 | Generating confident TEMR callsets and breakpoint annotations across human genomes. a** TEMR callset summary (dark slice) by TE type, SV type and TE orientation. TE, transposable element; SV, structural variant; MEI, mobile element insertion, TEMR, transposable element-mediated rearrangement; LINE; long interspersed nuclear element. **b** A diagram of TEMR structures showing distinct TEs (black and blue solid arrows) in the reference and a recombined TE in the sample (black and blue mixed arrows). The breakpoint junction in the sample is indicated by the dashed red line. Deletion and duplication TEs are largely mediated by TEs in the same orientation (solid green and purple), and all inversions are mediated by TEs in opposite orientation (hatched green and purple). **c** Median TEMR sizes differ by type of TE present at breakpoints across different SV types (*Alu* TEMRs vs LINE-1 TEMRs; deletion: $n = 361$ vs 84, duplication: $n = 29$ vs 4, and inversion: $n = 7$ vs 8). A two-sided Welch's *t*-test was used to calculate the p-value. n.s = not significant. **d** A 2,210 bp deletion TEMR between two *Alu* repeats in direct orientation. Top panel: UCSC genome browser image showing the deletion with breakpoints in an *Alu*Sp and an *Alu*Sg. Bottom panel: Breakpoint reconstruction of the assembled deletion (middle, NA19240) against *Alu* consensus sequences (top and bottom) identifies a 20 bp breakpoint microhomology (red). REF, reference genome (GRCh38).

breakpoint homology for deletions and duplications in our non-redundant TEMR callset using Manta, when available, and HiFi assembly callset otherwise. Since inversions had imprecise breakpoint junctions from SV callers, we used the manually curated junctions with breakpoint homology for the 15 inversions in our callset.

No single SV caller was able to precisely identify breakpoint junctions and junction homologies for all 70 TEMRs. When we extracted and inspected the DNA sequence from HiFi assemblies around the seven TEMRs with inaccurate breakpoint junctions (Methods), we were able to successfully identify the SNVs and indels that were causing inaccuracies. Due to limitations imposed by SRS sensitivity and methods to accurately place breakpoints with LRS, no single approach is able to detect and accurately characterize break-points for all TEMRs.

## Transposable elements mediate SVs by distinct mechanisms

To investigate the DNA repair mechanisms involved TEMR formation, we mapped TEs flanking TEMR events against their corresponding consensus sequences (AluY[54] and L1.3[53]), identified the position of 5′ and 3′ breakpoints, and annotated breakpoint homology (Fig. 2a) (Methods). We categorized TEMRs as products of homologous recombination (HR) if there was a significant overlap within homologous locations in the consensus repeat sequence and the overlap was identical to the breakpoint homology (Fig. 2b). Otherwise, they were categorized as products of non-homologous repair events (NHE; both end-joining and replication-based mechanisms) (Fig. 2c). We were able to systematically categorize 90.5% of the callset. The remaining 9.5% (47) of TEMRs that required manual inspection were comprised of events with breakpoints in the TE poly-A tail (long sequence of adenine nucleotides in plus strand/poly-T on minus strand) and truncated TEs.

Of all 493 TEMRs, 390 (79.1%) were categorized as HR (TEMR-HR, 354 Alu and 36 LINE-1) and 103 (20.9%) were categorized as NHE (TEMR-NHE, 43 Alu and 60 LINE-1) (Fig. 2d). We found that 89.2% of TEMR-HRs were driven by Alu elements and 62.5% of TEMR-NHEs were drive by LINE-1 elements. Furthermore, given the relative number of templates for homologous repair, most of the breaks that occur within an Alu element will likely be repaired with recombination with a nearby Alu element. Although Alu elements have far more homologous sub-strates, they comprise only half of the sequence content of the human genome compared to LINE-1 elements. Therefore, the likelihood of getting a random break in two LINE-1 elements followed by non-homologous repair is much higher than this occurring between Alu elements. Additionally, Alu elements (~300 bp) are composed of two ~150 bp homologous monomers, and we identified 10 direct Alu TEMRs with breakpoints in homologous regions of different monomers resulting in one chimeric Alu element with a single monomer (~150 bp) and nine chimeric Alu elements with 3 monomers (~450 bp) (Supplementary Fig. 4). While this imbalance appears to be significant ($p = 0.021$, two-tailed Binomial test), the mechanism responsible for a longer chimera preference is unclear.

We next examined the GC content of the microhomology greater than 5 bp at breakpoint junctions and found microhomologies of Alu TEMRs to be significantly enriched for GC content (median of 57.1%) compared to the Alu sequences in GRCh38 (median of 51.5%; $p < 0.01$, Welch's t-test) and the whole GRCh38 (median of 39.7%; $p < 1e^{-5}$, Welch's t-test) (Fig. 2e). We found no significant enrichment for GC content at the microhomology of LINE-1 TEMRs (median of 37.1%) compared to the LINE-1 sequences in GRCh38 (median of 33.5%). Higher GC content indicates a stronger strand annealing between G-C (three hydrogen bonds) compared to A-T (two hydrogen bonds) leads to more stability for repair intermediates, as was previously proposed[31]. Overall, the variation in the size of Alu and LINE-1 TEMRs and the divergence of G-C content at the breakpoints of HR events mediated by the two TEs may mean that the mechanisms governing

TEMRs can be dependent on the repeat class and/or size of the homology tract leading to stability of annealing prior to repair.

## Characteristics of TEMR breakpoints

Microhomologies were present at the junctions of 446 TEMRs varying from 1 bp to 307 bp with a median of 13 bp (Fig. 2d). TEMR-HRs have a median microhomology length of 17 bp and 93.3% of them have microhomologies 5 bp or longer. In contrast, TEMR-NHEs have a median microhomology length of 1 bp and nearly 96% of them have microhomologies shorter than 5 bp. Additionally, we observed the presence of small insertions ranging from 1 bp to 23 bp in 35% of TEMR-NHE events (median size of 2 bp), a known signature of end-joining repair mechanisms including micro-homology mediated end joining (MMEJ)[55]. We found Alu TEMRs have a shorter median microhomology length of 16 bp compared to 34 bp for LINE-1 TEMRs. We examined the Alu, and LINE-1 elements involved in TEMRs and found that 36 Alu subfamilies and 54 LINE-1 subfamilies were associated with TEMRs (Supplementary Table 4). We inspected the percent divergence among Alu and LINE-1 elements across the reference genome using the RepeatMasker dataset from UCSC genome browser and compared that to the Alu elements and LINE-1 elements from our TEMR callset. We found that Alu and LINE-1 elements from our callset have a lower median divergence when compared to the Alu and LINE-1 elements present within the latest reference genome (Alu: 9.6% vs 11.9%; $p < 1e^{-7}$, Welch's t-test, and LINE-1: 9.9% vs 21.6%; $p < 1e^{-7}$, Welch's t-test). Further, we found that AluS and AluY subfamilies were enriched within Alu TEMRs (AluY: 24.6% vs 11.8%, $p < 1e^{-22}$, two-tailed Fisher's exact test and AluS: 66.8% vs 57.4%, $p < 1e^{-7}$, two-tailed Fisher's exact test) and L1PA subfamilies were enriched within LINE-1 TEMRs compared to GRCh38 (53.6% vs 12.6%, $p < 1e^{-41}$, two-tailed Fisher's exact test) (Supplementary Table 5). This observation is in concordance with previous studies showing that younger TEs (fewer acquired mutations) are more likely to be involved in TEMRs[19], and AluS TEMRs are enriched due to their relative abundance (678,131 AluS compared to 139,234 AluY elements in GRCh38[3]).

We next investigated whether TEMR-HR events were more likely to be mediated by similar TEs; TEs involved in TEMR-HRs have a median similarity of 82.6% whereas TEMR-NHE events occurred between repeats that were significantly more diverged, with only 59% median similarity (Methods). This enrichment for similarity is a signature of recombination-mediated repair as opposed to non-homology-mediated mechanisms such as NHEJ. We found a significant difference in similarity between HR-driven and NHE-driven LINE-1 TEMRs (93.5% vs 47.9%, $p < 1e^{-5}$, Welch's t-test), and Alu TEMRs (82.2% vs 79.9%, $p < 0.01$, Welch's t-test) (Fig. 3a). Interestingly, we also observed the median difference between HR-driven and NHE-driven LINE-1 TEMRs was 20-fold higher than the difference between HR-driven and NHE-driven Alu TEMRs. These trends appeared to be consistent across variant types with no observable difference between deletions, duplications, and inversions for both LINE-1 and Alu events (Supplementary Table 6), although the number of duplications and inversions were too small for a test of significance.

We grouped TEMRs based on their mechanism (HR / NHE), family (Alu / LINE-1) and orientation of the TE involved (Direct / Indirect) and performed correlation analysis among three main characteristics used to discern TEMR mechanisms: the length of the TEMR, the tract length of homology at the breakpoint junction, and the similarity between the two TEs involved in the rearrangement (Supplementary Table 7). The homology length and TE similarity among TEMR-HRs were the only categories with statistically significant correlation value for both Alu TEMRs (correlation = 0.19, $p < 0.001$, Spearman's correlation) and LINE-1 TEMRs (correlation = 0.49, $p < 0.01$, Spearman's correlation). We also found that the percent similarity between the two Alu elements involved in an HR event is positively correlated (correlation = 0.13, $p = 0.018$, Spearman's correlation) with the size of TEMRs, but

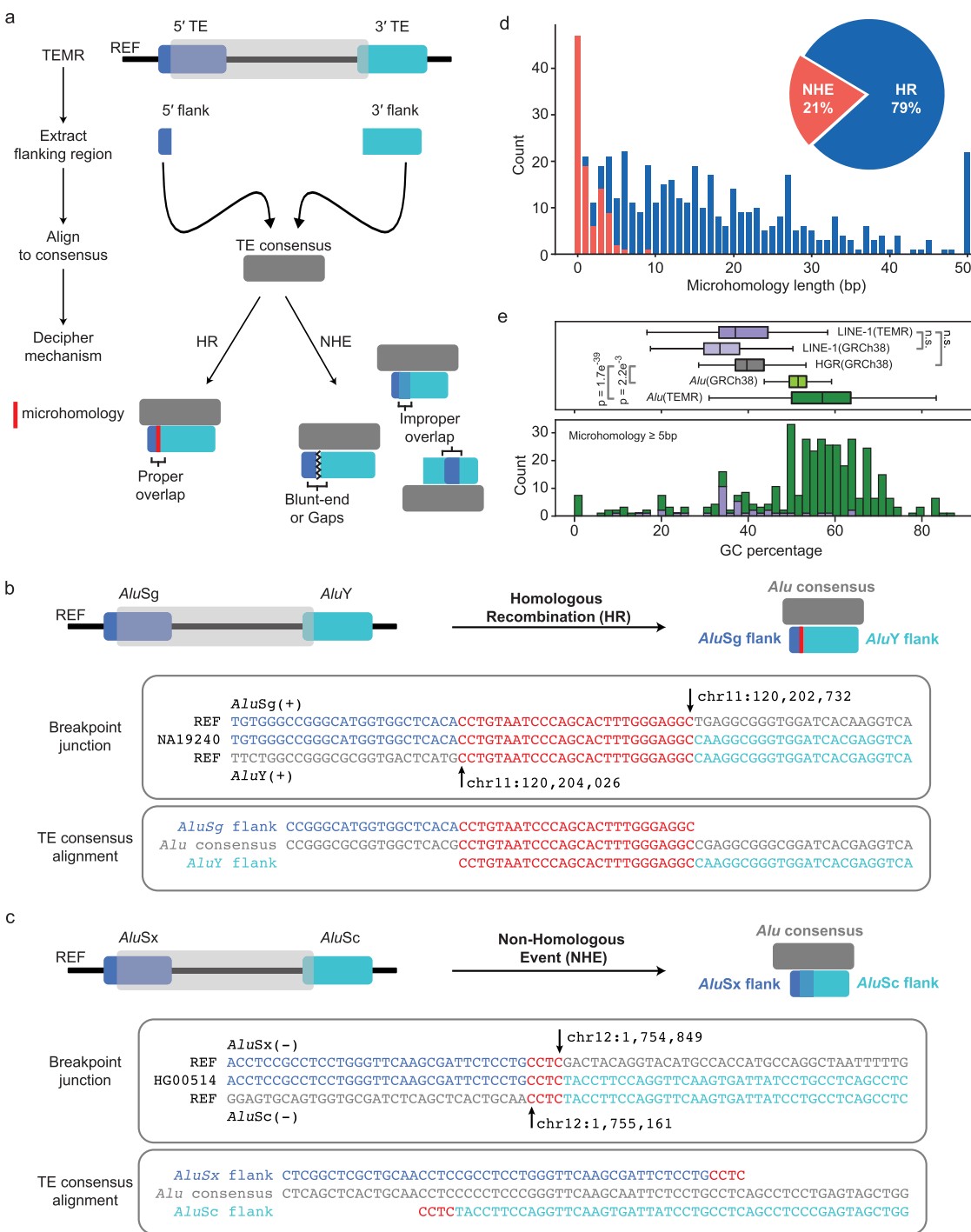

**Fig. 2 | Identifying mechanistic signatures of TEMRs. a** TEMR events are classified by breakpoint characteristics guided by TE consensus sequences. For homologous breakpoints, the chimeric TE resulting from the TEMR event must reconstruct a full TE with microhomology at the breakpoint. HR, homologous recombination; NHE, non-homologous repair; TE, transposable element; SV, structural variant; MEI, mobile element insertion, TEMR, transposable element-mediated rearrangement; LINE; long interspersed nuclear element. **b** An example of TEMR-HR (top). Breakpoint junction of 1,294 bp TEMR deletion in NA19240 (middle) and alignment between flanking *Alu* elements to a *Alu* consensus sequence (bottom). **c** An example of TEMR-NHE (top). Breakpoint junction of 312 bp TEMR deletion in HG00514 (middle) and alignment between flanking *Alu* elements to a *Alu* consensus sequence (bottom). REF, reference genome (GRCh38). **d** The breakpoint microhomology distribution differs between NHE (orange) and HR (blue) TEMRs. **e** top: Microhomology GC content distribution for *Alu* TEMRs (dark green, *n* = 330), reference *Alu* elements (light green, *n* = 1,181,072), LINE-1 TEMRs (dark purple, *n* = 38), reference LINE-1s (light purple, *n* = 962,085) and the full human genome reference (HGR, gray). Bottom: Average GC content of TEMR breakpoint microhomologies for *Alu* (green) and LINE-1 (purple) TEMRs. Microhomologies were restricted to 5+ bp for this analysis. A two-sided Welch's t-test was used to calculate the p-value. n.s, not significant.

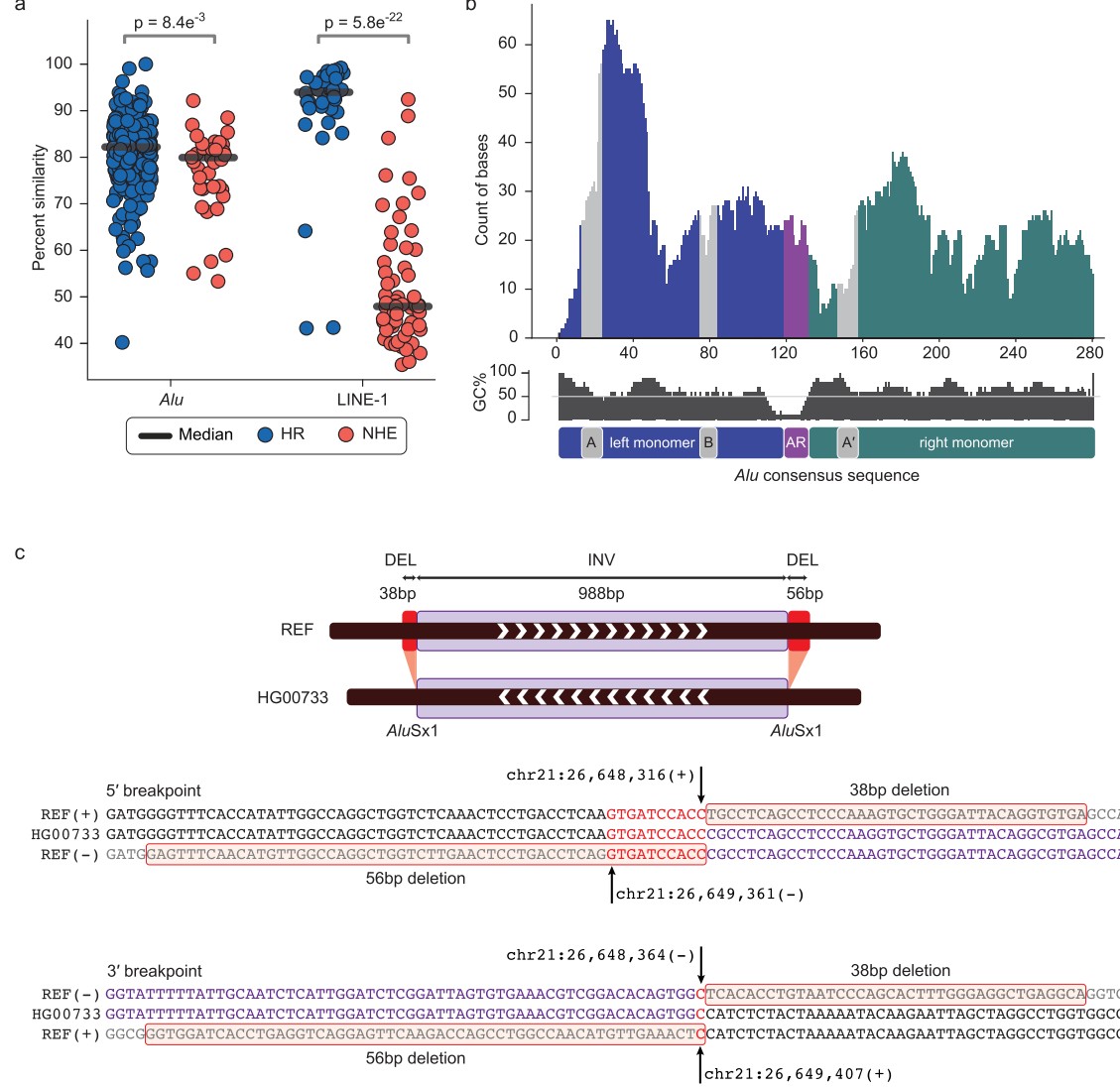

**Fig. 3 | Features and complexities of TEMRs. a** Median similarity between TEs at TEMR breakpoints differs by TE type and mechanism (TEMR-HR: 354 *Alu* and 36 LINE-1 and TEMR-NHE: 43 *Alu* and 60 LINE-1). A two-sided Welch's t-test was used to calculate the p-value. HR, homologous recombination; NHE, non-homologous repair; TE, transposable element; SV, structural variant; MEI, mobile element insertion, TEMR, transposable element-mediated rearrangement; LINE; long interspersed nuclear element. **b** Distribution of the breakpoint microhomology along the *Alu* consensus sequence. *Alu* elements consist of a left monomer (indigo), right monomer (green), RNA polymerase III promoter regions (gray A-Box, B-Box and A'-Box) and Adenosine Rich region (AR, purple). **c** Example of a 988 bp TEMR inversion (INV) with additional complex breakpoints (38 bp and 56 bp deletion (DEL) indicated in pink shaded sections) implicating replication-based mechanisms of variant formation. REF, reference genome (GRCh38).

this failed to meet our threshold for significance ($p < 0.01$). We were unable to find any other statistically significant (positive/negative) correlations among the examined features and believe this could be due to the small sample size, most notably of LINE-1 TEMRs. Additionally, we found that deletion and duplication TEMR-HRs were mediated by TEs in direct orientation, and that inverted orientation TEs were occasionally found at the junctions of NHE events. All inversions were mediated by TEs in opposite orientation and largely contain breakpoints consistent with HR (73%). We then plotted the microhomologies present at breakpoint junctions of 354 *Alu* TEMRs driven by recombination with respect to their relative position in an *Alu* consensus sequence[54] (Fig. 3b). Consistent with previous studies, we observe peaks near the 3' ends of the A-Box and A'-Box[19,31].

**Inversions are accompanied by complexities at the breakpoints**
Since inversions are known to harbor complex breakpoints junctions[56,57], we inspected the DNA sequence from HiFi assemblies for each of the 15 inversions in our final callset. Four of the seven *Alu*-mediated inversions and none of the eight LINE-1–mediated inversions contained additional complexities at both breakpoints. Complexities at *Alu*-mediated inversions included both deletions (38 bp to 251 bp) and insertions (14 bp to 23 bp) (Fig. 3c, Supplementary Fig. 5 and 6). Interestingly, we found that *Alu*-mediated inversions with complex breakpoints (median 1097 bp) were smaller than those TEMR inversions without complex breakpoints (median 4,372 bp). Furthermore, *Alu*-mediated inversions with homologous, simple breakpoints were shorter than LINE-1–mediated inversions (median 1646 bp vs 5,487 bp; $p < 1e^{-7}$, Welch's t-test). We also observed that the similarity between the two *Alu* elements (median 81.7%) involved in TEMR inversions were lower compared to similarity between the two LINE-1s (median 97.1%) involved. These breakpoint characteristics, percent similarity, and variant size for smaller *Alu*-mediated events reflect known repair mechanism signatures, such as MMBIR by break repair and serial replication slippage in cis by aberrant annealing within replication

bubbles[36,58], which contrasts with larger inversions mediated by NAHR[52].

## Complex rearrangements

While curating duplications determined using our ensemble pipeline with insertions from the HiFi assembly callset[16] (Methods), we found four TEMRs containing higher-order amplification (multiple copy number variant - mCNV) that were flanked by homologous TEs. One of these included a triplication within intron 1 of *DNASE1* (Supplementary Fig. 7). Additionally, we found a 6 kbp mCNV with a 1 bp breakpoint microhomology on chr17 that harbored a 2 kbp deletion (non-TEMR) with a 59 bp breakpoint microhomology within a copy (Fig. 4a–c). Due to high copy numbers involved, we manually curated these mCNV TEMR breakpoints from assembly data, estimated copy number using read-depth and performed ddPCR to orthogonally validate the copy number status for the region between TEs (Fig. 4d) (Methods). We found that read-depth-based approaches were able to accurately estimate the copy number status of events where the intervening region contained at least 7% of unique sequence. A single event was only amenable to ddPCR due to high repetitive content (54.4% *Alu* and 45.6% ERVL-MaLR) (Supplementary Fig. 8). We compared these mCNVs with all 64 haplotypes from the latest HGSVC publication[16] and found two mCNVs were present in other haplotypes (Supplementary Fig. 9). These finding are indicative of how TEMRs can facilitate additional copy gains after initial rearrangements.

## TEMRs lead to polymorphic insertions in human genomes

In addition to simple deletions, duplications, inversions, and complex rearrangements, TEMRs can lead to insertions of DNA[16,28]. These insertions are often representative of a polymorphic deletion relative to ancestral humans that became part of the reference by chance; therefore, the undeleted allele manifests as an insertion in SV callsets. Recently, a 4 kbp insertion in the first intron of the *LCT* gene was described where the deletion of the region (reference allele) could be responsible for adaptive evolution in the human population[16]. To better understand the role of TEMRs representing insertions of ancestral sequence, we compared the results of earlier studies[19,20] with the HiFi assembly data[16], and inferred the polymorphism rate of TEMR deletions between chimpanzee and human genomes (Supplementary Fig. 10a, b). By comparing the deletion in chimpanzees to the insertion in humans, we estimate that 20% of *Alu*- and 15% of LINE-1−mediated deletions were polymorphic in the human population (Supplementary Fig. 10c). This survey of ancestral regions inserted into human genomes by TEMRs also points to the role of TEs in genomic evolution and is probably a significant contributor to the role of TEs throughout speciation that is yet understudied.

## TEMRs contribute to variation in functional regions of human genomes

*Alu* elements, especially older subfamilies are known to cluster in genic, GC-rich regions of the genome[1,59–61], and may facilitate genic *Alu* TEMRs. To investigate this, we first inspected whether TEMRs occur more frequently within regions of high TE density, and then we inspected how frequently TEMRs occur within genic regions of the genome. We found that *Alu* TEMRs were enriched in *Alu* dense regions of the genome compared to other parts of the genome ($p < 1e^{-5}$, Welch's t-test) (Fig. 5a). The highest *Alu* density is observed on chromosomes 19 (25%) and 17 (18%)[28] compared to the whole genome (9.9%) (Methods). Further, we found the highest density of *Alu* TEMRs span callable regions for chromosomes 19 (40 events, 7.5 per 10 Mbp) and 17 (37 events, 5.1 per 10 Mbp) compared to the whole genome (397 events, 1.45 per 10 Mbp) (Methods). We also observed a similar pattern with LINE-1 TEMRs (96 events, 0.35 per 10 Mbp) in LINE-1 dense regions of the human genome ($p < 1e^{-5}$, Welch's t-test) (Supplementary Fig. 11).

To determine the effects of the 493 TEMRs identified in this study on genic regions, we intersected our 493 TEMRs with the RefSeq Gene database[62] using the Ensembl Variant Effect Predictor (VEP), which included curated and predicted set of genes[63]. We found that 5.7% (28) were exonic, 48.9% (241) were intronic, and an additional 10.5% (52) were within 5 kbp of a gene (Fig. 5b). Additionally, 95.2% (256 out of 269) of genic variants affect the curated set of protein coding genes (Supplementary Table 8). From TEMR deletions, we identified seven in 3′ UTR regions, nine in 5′ UTR regions, and 11 in coding sequence including 3 stop-loss events (Fig. 5c). TEMRs overlapped 126 cis-regulatory elements[64] and 70 transcription factor binding sites (TFBS)[63] enriched in gene-proximal sites (Fig. 5c). We noted that TEMRs compared to non-TEMRs are enriched in and surrounding genes (321 vs 2,740, $p < 0.001$, two-tailed Fisher's exact test), likely due to the higher intragenic concentration of *Alu* elements[65,66] (Supplementary Table 8). Further, we intersected 493 TEMRs with topologically associating domains (TADs) identified in NA12878[67] and found that 459 (83.1%) TEMRs were present completely within TADs and 1 TEMR was present at the edge of a TAD. One *Alu*-mediated deletion in the protein coding gene *TRIM65* eliminates a last exon, including a stop codon and the 3′ untranslated region (Fig. 5d). We verified that isoforms containing the deleted exon are expressed in a recent dataset comprised of 30 breast cancer samples[68]. Additionally, this deletion spans six regulatory elements annotated in the Open Regulatory Annotation dataset[69]. *TRIM65* is a well-known cancer biomarker and a potential therapeutic target for colorectal cancer and lung cancer treatment[70,71]. Further, we found 22 intragenic TEMRs overlapping hotspot OMIM (Online Mendelian Inheritance in Man) genes that are highly susceptible to *Alu*-driven genomic instability[31]. Our findings identify TEMRs as a prevalent reservoir of genic variation due to their distribution in the genome and reveal polymorphic SVs that may impact autosomal recessive disease-associated genes.

## Discussion

With nearly 4.5 million copies spread across the human genome, TEs are ideal substrates for rearrangements[1]. Due to the high sequence similarity between subfamilies of the same TE, it has been challenging to identify and characterize TEMRs across whole genomes, and overcoming both bioinformatic and sequencing hurdles was important to complete these analyses.

We identified 493 TEMRs in a genome-wide analysis of three genomes characterized with both long-read and short-read sequencing. Contrary to in vitro systems, which require varying the identity of the TEs in each assay, the characterization of rearrangements in human genomes provides a natural experimental framework for how TEMRs are formed. In general, we found that longer TEs mediate larger rearrangements, indicating that the length of homology might play a significant role in determining the mechanism of repair and the size of the resulting SV. Similarly, the orientation of flanking repeats, the similarity of those sequences, and the junction homology are all indicators of whether a TEMR occurred by HR or non-homologous repair (NHE). Similar TEs in direct orientation with longer homology at the junction are all indicative of HR. Conversely, TEs in inverted orientation with shorter junction homology points to NHE. Overall, 79.7% of TEMRs appear to be mediated by HR, yet the majority of HR-driven TEMRs have homology shorter than 40 bp. This is significantly shorter than the minimum processing segment required for nonallelic homologous recombination (NAHR) in mammalian genomes, where >100 bp of perfect sequence identity is needed[72]. Therefore, most of these events likely proceed in RAD51-independent mechanisms that benefit from annealing homologous sequences[35].

Recent studies have proposed single-strand annealing (SSA), microhomology-mediated break-induced replication (MMBIR), and microhomology-mediated end joining (MMEJ) as alternative mechanisms capable of driving TEMRs[28,31–35,73]. Due to the presence of

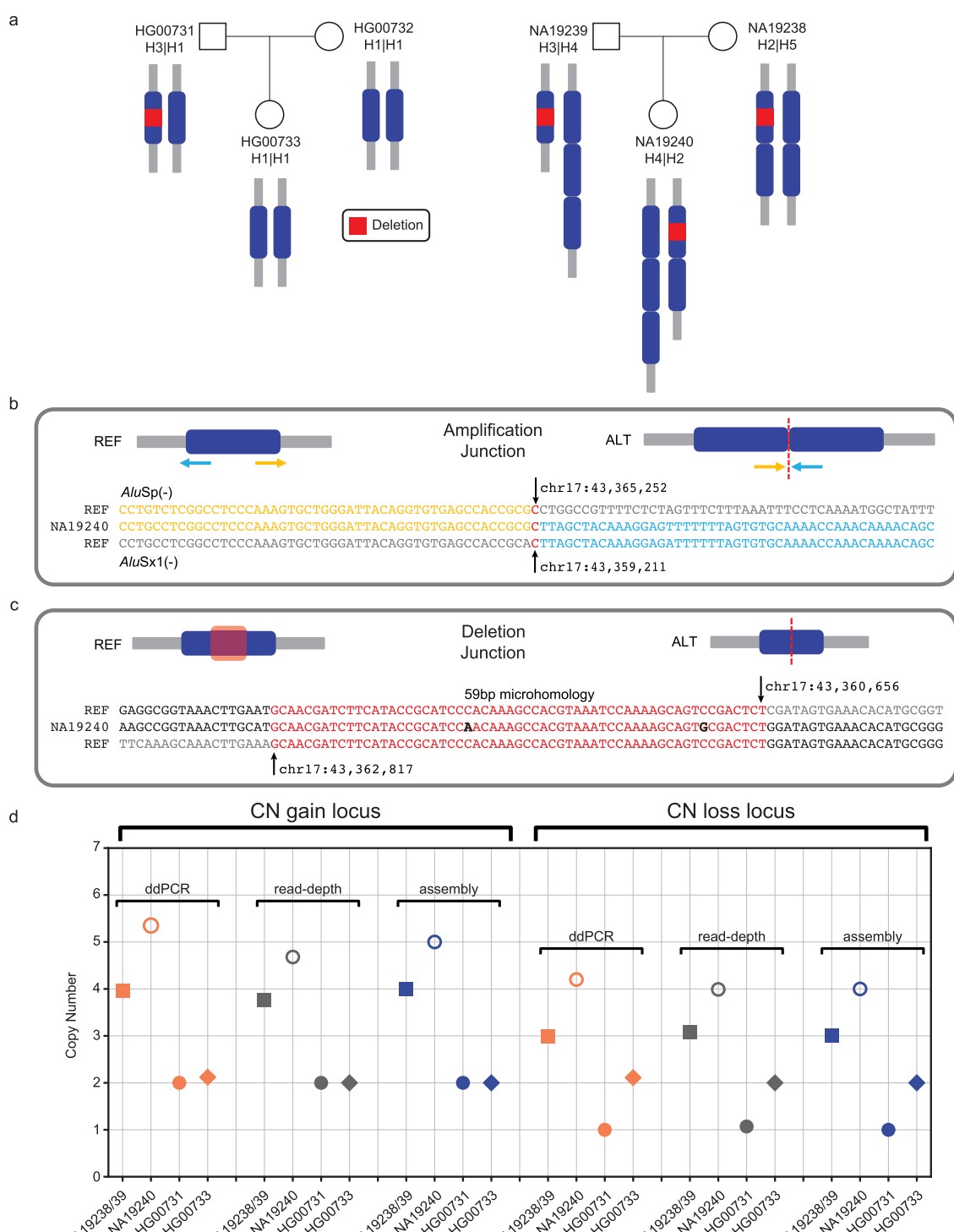

**Fig. 4 | TEs mediate multi-copy CNVs (mCNVs). a** 6 kbp multi-copy event (duplication and triplication) with a smaller 2.2 kbp deletion (red) in a subset of copies. **b** Reconstruction of the *Alu*S-mediated mCNV TEMR breakpoint surrounding each 6 kbp copy shows 1 bp microhomology (red). Arrows (blue and yellow) indicate the source of the sequence in reference (REF) and sample (ALT). The dashed red line is indicative of the breakpoint junction. **c** Reconstruction of the 2 kbp inner-copy deletion shows 59 bp of near-perfect homology (red bases). **d** Copy number status of the individuals containing the triplication found using ddPCR, read-depth analysis and assembly. TEMR, transposable element-mediated rearrangement; REF, reference genome (GRCh38).

homology at the junctions of TEMRs and the lower percent (~80%) of similarity between the TEs involved, it is likely that all these mechanisms play a role in their formation. The prevalence of shorter deletions in our study indicates a significant role for SSA or Alt-EJ (alternative-end joining) in the formation of deletion TEMRs. SSA is indicated by the homology of the repeats involved in ~80% of the rearrangements. If Alt-EJ was more common, shorter stretches of perfect

microhomology would potentially be more common, as would rearrangements between non-homologous regions of the *Alu* or LINE-1 elements. Alternatively, a hybrid approach that invokes both SSA and end joining has been proposed[33,35] with this mechanism, longer deletions could be accompanied by longer stretches of homology (imperfect homology[74]), as seen with the LINE-1 TEMRs in our study. In contrast to in vitro studies where more identity appears to be required

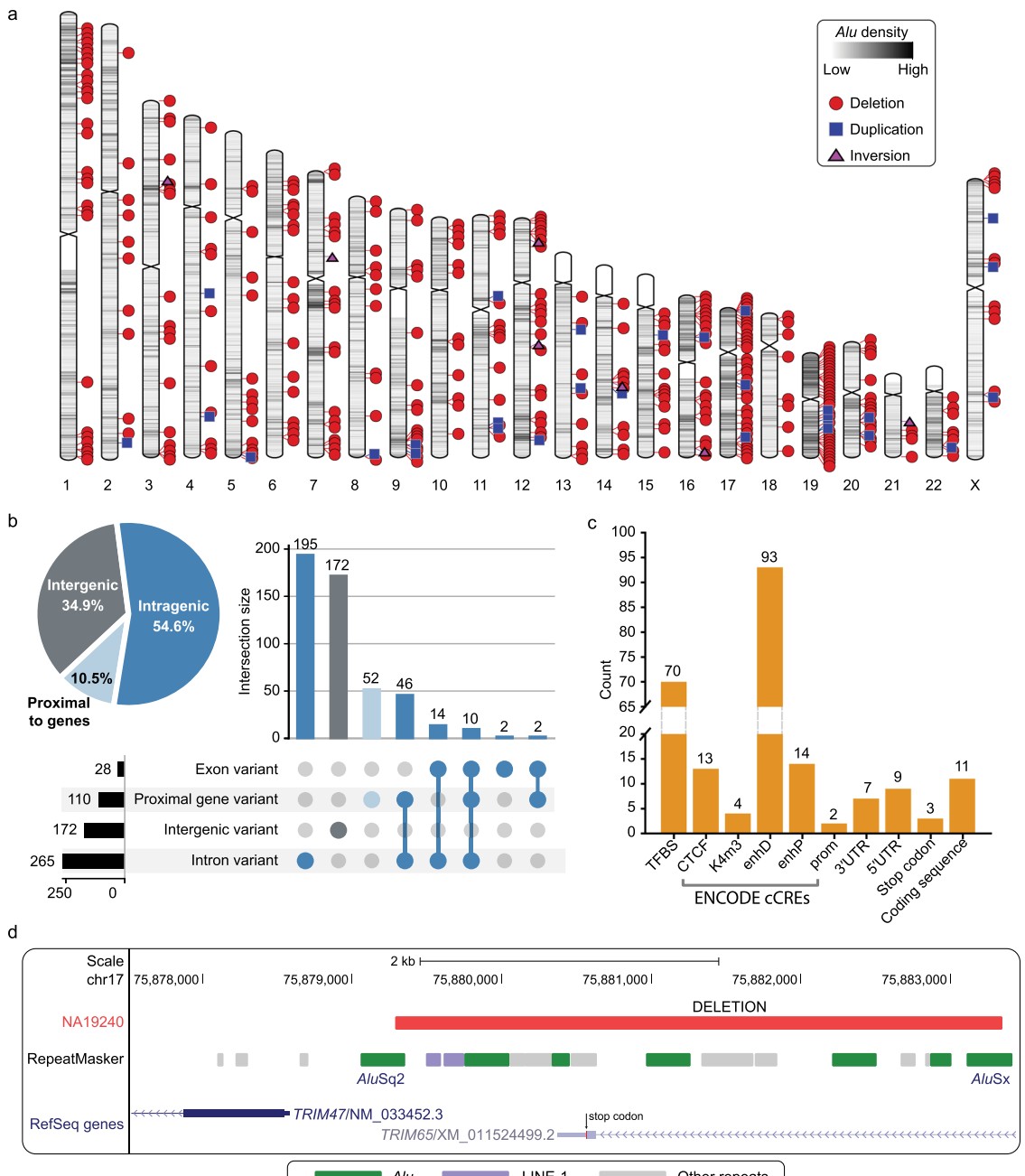

**Fig. 5 | TEMRs disproportionately affect genes. a** All 397 *Alu* TEMRs by variant type (shape) and *Alu* density (shading). **b** TEMRs disproportionately intersect genes (pie chart) affecting introns, exons, and gene-proximal regions that are often enriched with regulatory elements. **c** Regulatory and coding regions affected by TEMRs. Regulatory annotations were retrieved from ENCODE cCREs, coding regions and TFBS predictions from Ensembl Variant Effect Predictor (VEP). TFBS, transcription factor binding site; cCREs, Candidate Cis-Regulatory Elements; K4m3, DNase-H3K4me3; enhD, distal enhancer-like signature; enhP, proximal enhancer-like signature; prom, promoter-like signature; UTR, untranslated region; ENCODE, the encyclopedia of DNA elements; TEMR, transposable element-mediated rearrangement. **d** Example of TEMR deleting a stop codon in *TRIM65* (bottom).

for homology directed repair, the average divergence of TEs giving rise to a chimeric *Alu* at the TEMR junction was 80%. The overall distribution of junctions in the TEMR-HRs contains a peak near the 3′ ends of the A and A′ boxes (Fig. 3b), which could reflect preferential synapsis between GC-rich regions and *Alu* elements (Fig. 3c) or reflect the evolutionary constraint on the A and B boxes that yields more extensive homology tracts within these loci. Finally, the secondary structures inherent in *Alu* elements could lead to some regions being preferential sites of DNA repair.

We found significant sequence divergence between LINE-1s involved in TEMR-HR and TEMR-NHE but not between *Alu* elements

(Fig. 3a). Intriguingly, though we examined the three genomes for NHEJ events, they remained relatively rare (even between divergent repeats) comprising ~20% of the TEMRs. Nearly 60% of LINE-1 TEMRs were driven by NHE, whereas only 10% of *Alu* TEMRs were driven by NHE. The investigation of hundreds to thousands more of these events will enable insights that may be hard to gain from directed experimentation, including the binning of lengths of rearrangements, mechanisms, and divergence of repeats. Further genome analysis could uncover additional TE classes involved in rearrangements, including HERVs (human endogenous retroviruses)[75,76]. Finally, we can extend our analyses to investigate mechanistic signatures of repair

pathways, including enrichment of PRDM9 binding sites near the SV junctions and the formation of single nucleotide mutations and indels *in cis* with TEMRs[55].

Since approximately 0.3% of human genetic diseases have been associated with *Alu*-mediated SVs, we intersected our TEMRs with OMIM genes that had a risk score for *Alu* TEMRs that were greater than one[31]. We found 22 intragenic TEMRs overlapping these genes with a median allele frequency of 0.13. It is likely that TEMRs continue to be underestimated in human genetics due to the historic prevalence of array comparative genomic hybridization and short-read sequencing for genome-wide SV analyses. Population estimates of allele frequencies of TEMRs are needed to assess the true impact of this mechanism on phenotypically relevant genes. Further, polymorphic TEMRs can lead to evolutionary implications when they occur within genes or include regulatory sequences;[16] the impact of TEMRs can be relevant both between species and within species[16,19].

In three genomes, we annotated 493 TEMRs, including deletions, duplications, inversions, mCNVs, insertions (ancestral deletion), and complex rearrangements. The expansion of our analysis to further assembly-based SV callsets will broaden this pool of rearrangements and allow for the discovery of novel events, which is likely to uncover new biology. For example, in the current study, we found 10 TEMRs with breakpoint junctions in poly-A tails, suggesting that the poly-A tails of non-homologous repeats can potentially drive TEMRs. Additionally, we found an instance where the same *Alu* was involved in two different SVs (both an insertion and a duplication in humans). Interestingly, the junctions of both events bore the presence of the same 30 bp microhomology at the breakpoint, signifying hotspots within an element (Supplementary Fig. 12). Furthermore, we found that 30× coverage SRS is able to capture nearly 85% of TEMRs identified by 75× LRS data in this study (Supplementary Fig. 13), indicating that publicly available datasets could be used to understand the role of TEMRs across disease and population cohorts[16,45,46,73].

Overall, our results show that TEMRs are an important source of variation in human genomes that can arise from ectopic repair by distinct pathways, and can lead to diverse consequences. TEMRs mediate rearrangements spanning more than 500,000 bp in a human genome, and are therefore important to individual variation and to evolutionary processes; the investigation of additional high-quality assemblies will increase our understanding of the impact of transposable elements in mediating genomic rearrangements.

## Methods

### Pipeline for identifying, filtering, and merging SVs

**SV identification.** PCR-free Illumina short-read Whole Genome Sequencing (WGS) data (75× coverage) and PacBio continuous long-read (CLR) data (40× coverage) for three samples were downloaded from the HGSV consortium (Puerto Rican (PUR) HG00733; Southern Han Chinese (CHS) HG00514; and Yoruban (YRI) NA19240)[38]. Complete schematic for the short-read and long-read ensemble pipelines are shown in Supplementary Fig. 14. Raw paired-end sequencing data (FASTQ) were aligned against the human genome reference (GRCh38/hg38) using BWA MEM (v0.7.17)[77]. Similarly, long-read sequences (FASTQ) were extracted from the native PacBio files (bax.h5) using pbh5tools (v0.8.0; https://github.com/PacificBiosciences/pbh5tools) and aligned to the HGR using NGMLR (v0.2.6)[50]. The aligned files were converted (sam to bam), sorted, merged, and indexed using samtools (v1.7). SV calling was performed on the indexed short-read bam files using Manta (v1.3.2)[47], LUMPY (v0.2.13)[48], and DELLY (v.0.7.8)[49] and indexed long-read bam files using Sniffles (v1.0.7)[50], SVIM (v1.4.0)[51] and pbsv (v2.2.0; https://github.com/PacificBiosciences/pbsv). Read-depth information were annotated for the output generated by the three callers using Duphold (v0.2.1)[78]. Settings with which each caller was run can be found in Supplementary Table 9.

**SV filtering.** After generating SV calls, resultant VCF files were mined for the coordinates (chrom, chromStart, chromEnd), svType, caller name, paired-read (PR), split-read (SR) and read-depth (RD) for SVs with "FILTER = PASS", and this information was transferred to a tab separated file (TSV). For long-read SV calls, PR and SR were replaced with read-support (RS). We removed SVs within 500 bp of gaps and centromeres, and SVs that overlapped (50%) with simple repeats BEDTools *intersect* (v 2.29.2)[79]. We retained deletions, duplications, and inversions of size ranging from 50 bp to 50 kbp.

**SV merging.** After testing multiple approaches to merge (Supplementary Fig. 1a, b) and filtering SVs, we applied read-based (SR ≥ 5 or PR ≥ 10 for short-reads and RS ≥ 5 for long-reads) and depth-based (RD < 0.7 for deletions and RD > 1.3 for duplications) filters to our callset (Supplementary Fig. 1c, d). We merged and retained SVs from multiple callers using 80% reciprocal overlap (RO) with BEDTools *intersect* and a rank-based method.

**Rank-based merging.** We investigated the accuracy with which callers identified breakpoint junctions by comparing breakpoint junctions of deletion calls between our intersection callset (SVs identified by all 3 caller) and the truth set[38] using 80% RO. We calculated the deviation between the breakpoints at both 5′ and 3′ junctions. We ranked the three callers used in both short-read (1. Manta, 2. DELLY and 3. LUMPY) and long-read pipeline (1. pbsv, 2. Sniffles and 3. SVIM). When merging (80% RO) SVs in our ensemble callset, we retained the SV size and breakpoint data identified from the highest-ranking caller and removed SVs from other callers (Supplementary Fig. 15).

### Statistics

All statistical tests were performed with scipy (v1.5.0)[80].

### Breakpoint refinement

We overlapped our ensemble callset with HiFi assembly-based SV calls using 80% RO. Since duplications were also reported as insertions in the assembly callset, we followed a two-step approach to verify proper support for duplication from the assembly. First, we checked for insertions within our duplication (and 500 bp around the SV). Second, we compared the duplication size with the insertion size (90% match). We obtained the precise breakpoint junctions and microhomology information from the assembly callset[16] and annotated our TSV files accordingly. Further, we assessed the accuracy of the microhomologies by validating 70 SVs using PCR and Sanger sequencing.

### TE density

We computed density of TEs corresponding to the TEMRs (for example, *Alu* density was calculated for each *Alu* TEMRs) by calculating the percentage of TE sequence present in the 50 kbp encompassing the SV (25 kbp on either side, from the center of an SV). As a control, one million random 50 kbp regions containing at least two TEs of the same family were selected across the reference genome using BEDTools *shuffle*. We used Welch's unequal variance *t*-test from scipy (v1.5.0)[80] to calculate the statistical significance of the TE density between TEMRs and our control group. We also inspected the density of TEs within the callable regions of each chromosome to avoid any sequencing discrepancies. We considered regions of the genome as callable regions by excluding gaps, centromeres, segmental duplication (identity ≥ 95%) and long (≥ 5 kbp) tandem repeats.

### Percent similarity between two TEs

We extracted the TE sequence in positive orientation (TEs on the negative strand were extracted and the reverse compliment was generated to obtain the TE sequence) from the GRCh38 using Repeat-Masker v3.0[3] and BEDTools *getfasta*[79]. The percent similarity was

calculated by aligning the two sequences using swalign (v0.3.4) (https://github.com/mbreese/swalign) with the scoring matrix from EMBOSS Water[81].

## Extracting sequences from assembly data

We used subseq (v 1.0) (https://github.com/EichlerLab/seqtools) to extract raw sequence (SV + 500 bp window) of both haplotypes from the assembled data of the corresponding individuals[16]. BLAT was used to confirm the presence of an SV. We used the sequence containing the SV to manually reconstruct the breakpoint junction.

## Deciphering TEMR mechanisms

We obtained the TE sequences, location of the breakpoint junction, and the microhomology of all TEMRs (Supplementary Data 1–3). Then, we obtained the flanking regions outside the breakpoints and aligned it to a consensus TE (AluY[54] for Alu TEMRs and L1.3[53] for all LINE-1 TEMRs). We calculated the distance between the flanking regions (post alignment) along the consensus. We categorized TEMRs as HR-driven if the two flanking regions overlapped, and the overlap size is identical to the microhomology. For Alu TEMRs that failed in the above step and still had microhomology, we aligned them against each monomer in the consensus and repeated the process to identify TEMRs that recombined between different monomers (left monomer recombining with right). TEMRs that had no microhomology and flanking regions that failed to overlap along the consensus were categorized as NHE. TEMRs that were not characterized by the above two steps were manually curated, and a mechanism was assigned based on the formation of a chimera and length of homology at the junction.

## Copy number estimation for mCNV using read-depth

We used mosdepth (v0.3.2)[82] to calculate the average per-base depth across the mCNV loci. To calculate the copy number at the mCNV loci we divided the average per-base depth at the mCNV loci by the average per-base depth across the whole genome (genome coverage).

## Analyzing ancestral deletions from a previously published dataset

We download the Alu TEMRs[19] and LINE-1 TEMRs[20], and lifted over the coordinates to GRCh38 (LiftOver from UCSC genome browser) along with the corresponding deletion size in the chimpanzee genome. We then intersected these coordinates with the insertions from the latest HGSVC publication[16] using BEDTools window[79] (length of the consensus sequence was used as window size). We retained events if size of the deletion (from chimpanzee) and insertion (from human) had a 90% match.

## Validation of structural variants

**Sample Preparation.** Lymphoblastoid cell lines from three parent-child trios[38] were obtained from the Coriell Cell Repository (Catalog: HG00512, HG00513, HG00514, HG00731, HG00732, HG00733, NA19238, NA19239, NA19240) as part of the National Human Genome Research Institute (NHGRI) catalog (https://www.coriell.org/1/NHGRI). Briefly, lymphoblastoid cell lines were cultured in RPMI 1640 media (Thermo Fisher Scientific, 11875119) supplemented with Gibco™ Penicillin – Streptomycin (Thermo Fisher Scientific, 15-140-122) + 2 mM L-glutamine (Thermo Fisher Scientific, 25030149) and 15% Gibco™ fetal bovine serum (Thermo Fisher Scientific, 10-082-147) in T25, T75, and T150 flasks (USA Scientific, CC7682-4825 (T25), CC7682-4875 (T75), CC7682-4815 (T150)). Cells were triturated daily, and were passaged to new flasks at a density of ~250,000 – 500,000 cells/mL. Cells were frozen down in 50 mL conical tubes in aliquots of 35 million cells and were kept at −80 degrees prior to extraction of high molecular weight DNA with the Puregene blood DNA isolation kit and protocol (Qiagen, 158489).

**Designing primers for PCR and Sanger sequencing.** Primers were designed using the Primer3web interface (v4.1.0: https://primer3.ut.ee/). We obtained the DNA sequence of candidate SVs and their flanking regions (500 bp) using BEDTools getfasta[79] or UCSC Genome Browser (Get DNA in Window) and regions for primer pairs were identified within the flanking regions. To ensure high quality sequencing through the breakpoint, primers were designed to anneal to at least 75 bp from the predicted breakpoints. We required that at least one primer mapped (UCSC BLAT) to a unique region in the reference genome (GRCh38) and the primer pair uniquely amplified the locus of interest (UCSC In-Silico PCR).

**PCR reaction.** PCR was conducted using the AccuPrime Taq DNA polymerase system (Thermo Fisher Scientific, 12339024). The following conditions were used for 25 µl PCR reactions:

- 2.5 µl of AccuPrime 10× Buffer II
- 1.5 µl of 10 µM forward and reverse primers
- 3 µl of 5 M Betaine
- 1 µl of DNA (20 ng/µl)
- 0.25 µl of AccuPrime Taq DNA polymerase
- 16.75 µl of nuclease free water

We utilized touchdown PCR cycling for optimal primer annealing and to achieve specific amplification of the desired region. The thermocycler program was as follows:

Step 1: 94 °C for 2 min
Step 2: 94 °C for 30 s
Step 3: 63 °C for 30 s (with a 1 °C ramp down per cycle)
Step 4: 68 °C for 1 min
Step 5: Return to step 2 and repeat for 8 cycles
Step 6: 94 °C for 30 s
Step 7: 57 °C for 30 s
Step 8: 68 °C for 1 min
Step 9: Return to step 6 and repeat for 25 cycles
Step 10: 68 °C for 1 min
Step 11: 4 °C infinite hold

PCR products combined with 5 µl of 10× OrangeG gel loading dye (0.4% w/v final concentration of OrangeG sodium salt, 40% w/v final concentration of sucrose, and sterile water) were run in 1% agarose gels with 0.1 µl ethidium bromide (10 mg/ml Bio-Rad, 1610433) per milliliter of gel. 7 µl of a 1Kb Plus DNA ladder (Thermo Fisher Scientific, 10787018) diluted in 10× BlueJuice Gel Loading Buffer (Thermo Fisher Scientific, 10816015) was run alongside the reactions. Bands at the desired size were excised under a blue light and DNA was purified using the Zymoclean Gel DNA Recovery Kit (Zymo Research, D4008).

**Sanger sequencing.** Purified PCR products were subjected to Sanger dideoxy sequencing (Eton Biosciences) along with their respective PCR primer pairs. If sequencing using PCR primers was expected to cause premature polymerase slippage due to a homopolymeric region being present between the primer and the breakpoint, separate sequencing primers were designed to avoid such regions.

**Droplet digital PCR (ddPCR) assay and probe design.** Custom primers and probes were designed for each mCNV using Primer3Plus (https://www.bioinformatics.nl/cgi-bin/primer3plus/primer3plus.cgi). Primers and probe assays were designed using the parameters recommended by Bio-Rad's ddPCR application guide. We selected primer pairs that had at least one primer mapping (UCSC BLAT) to a unique region in the reference genome (GRCh38) and generated a shorter (<200 bp) product. The reference probe and assay used for the duplexed reaction was designed to diploid gene RPP30. A HEX fluorophore (Bio-Rad, 10031279) was used for the reference assay and FAM fluorophore for the target assay (Bio-Rad, 10031276). We used a 3.6:1 nM primer to probe concentration ratio. Restriction enzymes

(Supplementary Table 10) from New England Biolabs (NEB) were used to fragment tandem duplications outside of predicted PCR product.

ddPCR Sample Preparation and Thermocycler Program: ddPCR reaction mix was assembled according ddPCR™ Copy Number Variation Assays (10000050421 Ver A) from Bio-Rad. The following concentrations were used for 22 µl of total volume prior to droplet formation:

- 11 µl of 2× ddPCR Supermix for Probes (No dUTP)
- 1.1 µl of 20× target assay
- 1.1 µl of 20× reference assay
- 1.5 µl of digested gDNA (20 ng/µl)
- 7.3 µl of nuclease free water

Droplets were formed by Bio-Rad's QX200 AutoDG Droplet Digital PCR system. After sealing the plates, droplet samples were loaded in a thermocycler.

Thermocycler program for 40 µl volume:
Step 1: 95 °C for 10 min
Step 2: 94 °C for 30 s
Step 3: 60 °C for 1 min (with a 2 °C/s ramp per cycle)
Step 4: Return to step 2 and repeat for 39 cycles
Step 5: 98 °C for 10 min
Step 5: 4 °C infinite hold

Plates were loaded into the QX200 Droplet Reader post PCR and copy number status was analyzed using QuantaSoft™ software from Bio-Rad.

### Reporting summary
Further information on research design is available in the Nature Portfolio Reporting Summary linked to this article.

## Data availability
Illumina SRS and PacBio LRS data for the samples used in this study were downloaded from publicly available database at International Genome Sample Resource (IGSR) at https://www.internationalgenome.org/data-portal/data-collection/structural-variation;[38] https://www.internationalgenome.org/data-portal/data-collection/hgsvc2[16]. The human reference genome GRCh38/hg38 [https://hgdownload.soe.ucsc.edu/goldenPath/hg38/chromosomes/] was downloaded from UCSC Genome Browser. NCBI Refseq dataset was downloaded from https://www.ncbi.nlm.nih.gov/projects/genome/guide/human/index.shtml. L1 recombination associated deletion[20] data were download from https://biosci-batzerlab.biology.lsu.edu/supplementary_data/Han_et_al_L1RAD_SI_Table_S4.doc. Alu recombination-mediation deletion[19] data was downloaded from https://biosci-batzerlab.biology.lsu.edu/supplementary_data/Sen_et_al_Suppl_Data.zip. SV and TEMR data files have been deposited in Zenodo with the following accession code: https://doi.org/10.5281/zenodo.7272154. Primers used for PCR, Sanger sequencing, and ddPCR can be found in supplementary information file. Source data are provided with this paper.

## Code availability
The code for the current version of the integrated pipeline used to analyze SV calls and identify TE-mediated rearrangements and any future version will be available at https://github.com/parithi-b/TEMR_analysis_pipeline. Data and codes used in this study have also been deposited in Zenodo with the following accession code: https://doi.org/10.5281/zenodo.7272154.

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

## Acknowledgements

We thank the members of the Beck lab for reading and editing the manuscript, in particular Alex V. Nesta and Ardian Ferraj. We also thank members of the Charles Lee lab and Miriam Konkel for their critical review of the manuscript. We thank Guruprasad Ananda for his advice on SV filtering and merging and Denisse Tafur for her help procuring and growing cell line. We would also like to thank the Cellular Engineering group at The Jackson Laboratory, especially Mallory Ryan and Sofia Giansiracusa, for their help with ddPCR experiments. This work was supported in part by the National Institutes of Health grants R00GM120453 and R35GM133600 from the NIGMS, P30CA034196 from the NCI, and startup funds from the University of Connecticut Health and The Jackson Laboratory to Christine R. Beck.

## Author contributions

C.R.B. conceived of and supervised the study. P.B. performed data preprocessing, SV calling, quality control, method development, data analysis and data interpretation. P.B., I.A.W., J.I.F., and J.N.D. designed primers for validation work. I.A.W., J.I.F., and J.N.D. performed PCRs and Sanger sequencing. P.B. and I.A.W. performed junction reconstruction work. J.I.F. designed and performed ddPCR experiment. P.A.A. provided homology data for method development. P.B. and C.R.B. wrote the manuscript with input from all coauthors. All contributing authors have read, edited, and approved the manuscript.

## Competing interests

The authors declare no competing interests.
