## [Peer Review File · Nature Communications]

Transposable element-mediated rearrangements are prevalent in human genomesREVIEWER COMMENTS

Reviewer #1 (Remarks to the Author):

This manuscript reports a very thorough analysis of transposable element-mediated rearrangements (TEMRs) based on both short-read and PacBio long-read whole-genome sequencing of three individuals (each of different genetic ancestry). The methods are appropriate, and the results are clearly described. I have just a few questions and comments.

Line 151: The term “non-TEMR” is not defined, though I would assume it refers to rearrangements that are not mediated by TEs. This should be clarified.

Line 164: It makes sense that TEMRs mediated by full-length LINE-1 elements are larger than those mediated by Alus because of size differences in the two types of TEs. Most LINE-1 elements, however, are truncated to 1 kb or so; thus, it would be predicted that TEMRs mediated by truncated LINE-1s should be shorter than those mediated by full-length LINE-1s, but shorter than those mediated by Alus. Were truncated LINE-1s examined, and is this the case?

Line 211: It’s interesting that the great majority of TEMRs that involved homology-directed repair are Alu-mediated (90%), while the majority that involve non-homologous end joining (60%) are mediated by LINE-1s. Given the sample size, this difference is almost certain to be statistically significant. The authors should discuss some reasons why this difference is seen.

Three types of TEs are active in humans: Alu, LINE-1, and SVA. Although SVAs are much less prevalent than the other two elements, it would be interesting to know whether they were also considered (or the authors could explain why they were not considered).

Reviewer #2 (Remarks to the Author):

Interesting and well-conducted research shows the key role of transposable elements (TE) in mediating genomic rearrangements. By analysing the three human genomes, 493 TE mediated rearrangements (TEMRs) were identified using both long read and short read sequencing. For randomly selected 70 TEMRs the precise junctions have been ascertained using PCR and Sanger sequencing. For all rearrangements their features (size, homology, orientation, GC content, TE density, etc) were

investigated and compared between different groups by means of statistical tests (mostly Welch's t-test). Moreover, the mechanism behind TEMRs formation has been deciphered for most cases.

The methodology for TEMRs detection and statistical analysis is sound and robust. Several interesting findings are presented and hypotheses about the impact of TEMRs on evolutionary processes are stated. The paper is well written and the presentation is clear. The study is of interest for a wide audience and is worth publishing, but the following additional analyses would greatly improve the impact and clarity of the results.

major:

(i) The number of statistical tests performed revealed some rules that guided the formation of TEMRs (like longer TE mediate larger rearrangements). It would be great to propose a more comprehensive model explaining this phenomena with the use of nominal logistic regression or Poisson regression.

(ii) Beside the contribution of TEMRs to variation in functional regions, the impact on position effects for rearrangements spanning non-coding regions should be quantified by the analysis of chromatin conformation and the interactions between gene promoters and their regulatory elements.

(iii - optional) All TEMRs detected in the study of three genomes are mediated by Alu and LINE-1 elements, which are ubiquitous in the human DNA. The interesting open question is whether other kinds of transposable elements also mediate rearrangements not necessary by non-allelic homologous recombination mechanisms. Such targeted analysis should cover more genomes, but finding a rearrangement should be feasible if we focus on specific elements (e.g. HERVs) with well-defined parameters.

minor:

line 129: Provide the arguments the choice of three specific genomes.

line 240: Is overrepresentation of younger TEs statistically significant ?

Figure 1: make sure you are using a colour blind palette.

Figure 2: make the font for panels b. and c. the same size as in Figure 1 panel d.

Figure 4: consider using different points symbols for different genomes in panel d.

Reviewer #3 (Remarks to the Author):

This is a review of NCOMMS-22-16442-T, "Transposable element-mediated rearrangements are prevalent in human genomes." Herein, the authors use computational methods with deep sequencing data to identify structural variants (SVs) that have transposable elements (TEs) at the junctions in three human individuals, and then perform a series of analyses on these SVs, including some validation experiments with PCR/Sanger sequencing, and describing features of the SVs, such as homology length, degrees of sequence divergence, and sizes of the SVs. Among the 493 TE-associated SVs, most of them were deletions, but some were more complex, and an analysis of these complex events are shown, along with analysis of the propensity for these rearrangements in gene bodies. I found this analysis quite compelling, and its impact in part will be as a guide for mechanistic studies to determine the DNA repair pathways that mediate these events.

Major comments:

1. I suggest that the impact of the study could be enhanced by analyzing whether certain features of TE-mediated SVs show correlations, or not. Namely, perform correlation analysis with these parameters they describe: homology / non-homology and also homology length, vs. SV size, and vs. TE divergence. Does homology length, or % homology vs. non-homology correlate with SV size? With TE divergence? Does SV size correlate (positively or negatively) with TE divergence? Since the mechanism of repeat-mediated SVs can be affected by repeat distance and divergence (e.g. PMID: 32023454), such analysis would be very useful to the field. Of course, the lack of a correlation might be due to small sample size, but that caveat could be included in the text.
2. For TE divergence, is it possible to relate this to overall degrees of TE divergence for different types of TEs in the genome? Namely, some context for the degrees of TE divergence could enhance the study.
3. The initial section of the results could be expanded. In particular, a narrative to explain what was particularly innovative about this informatics pipeline to identify TE-mediated SVs would improve the accessibility of the study.

Reviewer comments

Reviewer #1 (Remarks to the Author):

This manuscript reports a very thorough analysis of transposable element-mediated rearrangements (TEMRs) based on both short-read and PacBio long-read whole-genome sequencing of three individuals (each of different genetic ancestry). The methods are appropriate, and the results are clearly described. I have just a few questions and comments.

We thank the Reviewer for their appreciation of our manuscript and the appropriateness of the methods and data interpretation.

Line 151: The term “non-TEMR” is not defined, though I would assume it refers to rearrangements that are not mediated by TEs. This should be clarified.

To clarify this point, we have amended the text where TEMRs are first introduced to read *“SVs with both breakpoints in different TEs of the same element class were categorized as TEMRs (Methods). In contrast, SVs with zero or one breakpoint within a TE, or with both breakpoints within different types of TEs were classified as non-TEMR events.”*

Line 164: It makes sense that TEMRs mediated by full-length LINE-1 elements are larger than those mediated by Alus because of size differences in the two types of TEs. Most LINE-1 elements, however, are truncated to 1 kb or so; thus, it would be predicted that TEMRs mediated by truncated LINE-1s should be shorter than those mediated by full-length LINE-1s, but shorter than those mediated by Alus. Were truncated LINE-1s examined, and is this the case?

We thank the Reviewer for this insightful question; we did indeed consider both full-length and truncated LINE-1s for this study. We performed the requested analysis, which had the expected result, and have updated the text to reflect the data: *“We found that Alu TEMRs (median length of 1,163 bp) are typically shorter than LINE-1 TEMRs³² (median length of 4,469 bp; $p < 1e^{-5}$, Welch’s t-test); this includes both full-length (7,663 bp; $p < 1e^{-5}$, Welch’s t-test) and truncated LINE-1 elements (median length of 3,618 bp; $p < 1e^{-4}$, Welch’s t-test) (Fig. 1c).”*

Line 211: It’s interesting that the great majority of TEMRs that involved homology-directed repair are Alu-mediated (90%), while the majority that involve non-homologous end joining (60%) are mediated by LINE-1s. Given the sample size, this difference is almost certain to be statistically significant. The authors should discuss some reasons why this difference is seen.

The discrepancy between Alu and L1 for NHE events is indeed significantly different ($p < 1e^{-23}$, two-tailed Fisher’s exact test). We have added this point in the results: *“We found that 89.2% of TEMR-HRs were driven by Alu elements and 62.5% of TEMR-NHEs were driven by LINE-1 elements.”*

Furthermore, we expect that given the relative number of templates for homologous repair, most of the breaks that occur within an Alu will be repaired by homology-mediated processes between the element with a break and a nearby Alu. However, although Alu elements have far more neighboring homologous substrates, they comprise only ~1/2 of the sequence content of the human genome that LINE-1 sequences do. Therefore, the likelihood of getting a random break in two LINE-1 elements followed by NHEJ is much higher than Alu sequences. We have added this point to the discussion and the sentence now reads *“Furthermore, given the relative number of templates for homologous repair, most of the breaks that occur within an Alu element will likely be repaired with recombination with a nearby Alu element. Although Alu elements have far more homologous substrates, they comprise only half of the sequence content of the human genome compared to LINE-1 elements. Therefore, the likelihood of getting a random break in two LINE-1 elements followed by non-homologous repair is much higher than this occurring between Alu elements.”*

Three types of TEs are active in humans: Alu, LINE-1, and SVA. Although SVAs are much less prevalent than the other two elements, it would be interesting to know whether they were also considered (or the authors could explain why they were not considered).

We did consider other types of TEs when identifying TEMRs, however, due to the low number of these events and difficulties aligning them to a consensus sequence we initially decided to exclude them. We have now updated the results and added a supplementary table (Supplementary Table 1) with the size information of each type. The updated text now reads: *“From our high-confidence callset of 5,297 SVs, we identified 543 nonredundant TEMRs (10.25%) across all three individuals (Fig. 1a). We identified an average of 263 TEMRs per sample (236 from PUR, 236 from CHS, and 316 from YRI) and they collectively affected an average of 795 kbp per sample. The 543 TEMRs consisted of 11 classes of TEs: Alu (397), LINE-1 (96), ERVL-MaLR (14), ERV1 (11), ERVL (8), L2 (6), ERVK (3), MIR (2), SVA (2), TcMar-Mariner (2), TcMar-Tigger (1), and hAT-Charlie (1) (Supplementary Table 1). Due to the prevalence of LINE-1 and Alu-mediated events, the difficulties in aligning ERVs and divergent transposons to consensus sequences, and the small number of TEMRs driven by non-Alu or LINE-1 categories precluding extensive mechanistic work, we focused on the two primary categories of TEMR in this study (493: 397 Alu and 96 LINE-1).”*

We also updated Fig. 1a with an additional category called “Other TEs” which contains non- Alu or LINE-1 TEMRs.

We manually inspected the two SVA-driven TEMRs and found them to be HR driven:

CHR	POS	END	SVTYPE	SVLEN	HOMLEN	TE_5'	TE_3'	ORIENTATION
chr19	20148842	20152102	DEL	3260	256	SVA_F	SVA_D	SAME
chr4	150954410	150964467	DEL	10057	320	SVA_B	SVA_B	SAME

Additionally, we inspected the other 48 TEMRs (those driven by TEs other than *Alu*, LINE-1 and SVA) and found the median homology length at the breakpoint junction to be 4 bp and median size to be 795 bp. Due to the difficulties that preclude extensive mechanistic work with these classes of TEMRs, we added them to Supplemental Table 1, but did not investigate the mechanisms or consequences of these events.

Reviewer #2 (Remarks to the Author):

Interesting and well-conducted research shows the key role of transposable elements (TE) in mediating genomic rearrangements. By analysing the three human genomes, 493 TE mediated rearrangements (TEMRs) were identified using both long read and short read sequencing. For randomly selected 70 TEMRs the precise junctions have been ascertained using PCR and Sanger sequencing. For all rearrangements their features (size, homology, orientation, GC content, TE density, etc) were investigated and compared between different groups by means of statistical tests (mostly Welch's t-test). Moreover, the mechanism behind TEMRs formation has been deciphered for most cases.

The methodology for TEMRs detection and statistical analysis is sound and robust. Several interesting findings are presented and hypotheses about the impact of TEMRs on evolutionary processes are stated. The paper is well written and the presentation is clear. The study is of interest for a wide audience and is worth publishing, but the following additional analyses would greatly improve the impact and clarity of the results.

We thank the Reviewer for their kind words about our manuscript, insightful evaluation of our analysis and findings, and recognition of the important role of transposable elements in the formation of structural variants.

major:

(i) The number of statistical tests performed revealed some rules that guided the formation of TEMRs (like longer TE mediate larger rearrangements). It would be great to propose a more comprehensive model explaining this phenomena with the use of nominal logistic regression or Poisson regression.

We appreciate the potential benefit of applying regression models to our callset. We interrogated seven features: percent similarity, homology length, TEMR size, 5' TE size, 3' TE size, GC percentage of 5' TE, and GC percentage of 3' TE with a logistic regression model using (with KFold cross validation; k=10) the sklearn package in python.

We obtained the estimated coefficients for the features used with the following scores: accuracy of 99% on training dataset, 94% on the test dataset, precision of 96%, recall of 96.5%, and F1 score of 92.6%.

Features	SIMILARITY	HOMLEN	SIZE	5PRIME_SIZE	3PRIME_SIZE	5PRIME_GC	3PRIME_GC
Coeff	0.0029	-1.33	0.0000087	0.0001	-0.0003	-0.24	-0.14

Certain criteria for the features used in this study were clearly linked with the mechanisms because of prior knowledge, such as higher homology length in HR-driven, indirect TEs driving NHE events. We believe with the given features used in this study and the sample size we are unable to fit a regression model to explain the different mechanisms.

Additionally, we performed correlation analysis between the features used to study TEMRs in this manuscript and the mechanisms of rearrangement. We added the following paragraph to the result “We grouped TEMRs based on their mechanism (HR / NHE), family (Alu / LINE-1) and orientation of the TE involved (Direct / Indirect) and performed correlation analysis among three main characteristics used to discern TEMR mechanisms: the length of the TEMR, the tract length of homology at the breakpoint junction, and the similarity between the two TEs involved in the rearrangement (Supplementary Table 7). The homology length and TE similarity among TEMR-HRs were the only categories with statistically significant correlation value for both Alu TEMRs (correlation = 0.19, $p < 0.001$, Spearman’s correlation) and LINE-1 TEMRs (correlation = 0.49, $p < 0.01$, Spearman’s correlation). We were unable to find any other statistically significant (positive/negative) correlations among the examined features and believe this could be due to the small sample size, most notably of LINE-1 TEMRs.”

The below table has been added to the supplementary material to summarize the correlation analysis (Supplementary Table 7). The highlighted rows indicate the features and categories that had a statistically significant result ($p < 0.01$).

Mechanism	TE	Orientation	Feature_1	Feature_2	r-value (spearman)	p-value (spearman)
HR	Alu	SAME	HOMLEN	PCT_SIMILARITY	0.1932	0.00026
			HOMLEN	SV_LENGTH	0.08868	0.09573
			PCT_SIMILARITY	SV_LENGTH	0.12577	0.01791
HR	LINE-1	SAME	HOMLEN	PCT_SIMILARITY	0.49028	0.0024
			HOMLEN	SV_LENGTH	-0.26838	0.1135
			PCT_SIMILARITY	SV_LENGTH	-0.24659	0.14711
NHE	Alu	SAME	HOMLEN	PCT_SIMILARITY	-0.34638	0.1143
			HOMLEN	SV_LENGTH	0.11569	0.60818
			PCT_SIMILARITY	SV_LENGTH	0.05251	0.81647
NHE	Alu	OPP	HOMLEN	PCT_SIMILARITY	0.23634	0.36111
			HOMLEN	SV_LENGTH	0.11883	0.64966
			PCT_SIMILARITY	SV_LENGTH	-0.2451	0.34305
NHE	LINE-1	SAME	HOMLEN	PCT_SIMILARITY	0.02663	0.88114
			HOMLEN	SV_LENGTH	0.13095	0.4604
			PCT_SIMILARITY	SV_LENGTH	0.29015	0.096
NHE	LINE-1	OPP	HOMLEN	PCT_SIMILARITY	0.02352	0.90922
			HOMLEN	SV_LENGTH	0.1013	0.62244
			PCT_SIMILARITY	SV_LENGTH	-0.32923	0.10052

Supplementary Table 7. Summary of the correlation analysis.

(ii) Beside the contribution of TEMRs to variation in functional regions, the impact on position effects for rearrangements spanning non-coding regions should be quantified by

the analysis of chromatin conformation and the interactions between gene promoters and their regulatory elements.

Since TADs are well conserved within and across species, we decided to utilize GM12878, a well characterized lymphoblastoid genome, for this analysis. We intersected 493 TEMRs with TADs identified in GM12878 (PMID: 25497547) and updated the manuscript accordingly: *“Further, we intersected 493 TEMRs with topologically associating domains (TADs) identified in GM12878⁶⁷ and found that 459 (83.1%) TEMRs were present completely within TADs and 1 TEMR was present at the edge of a TAD”*

(iii - optional) All TEMRs detected in the study of three genomes are mediated by Alu and LINE-1 elements, which are ubiquitous in the human DNA. The interesting open question is whether other kinds of transposable elements also mediate rearrangements not necessary by non-allelic homologous recombination mechanisms. Such targeted analysis should cover more genomes, but finding a rearrangement should be feasible if we focus on specific elements (e.g. HERVs) with well-defined parameters.

In fact, we did consider other types of TEs when identifying TEMRs, however, due to smaller sample size and difficulties in aligning them to consensus sequences, we decided to exclude them. We do agree that with a larger TEMR sample size we could potentially uncover other TEs driving rearrangements, but this will require more analysis of additional genomes and extensive manual curation (with current techniques).

Based on this question we have updated the results and added a supplementary table (Supplementary Table 1) with the size information of each additional TEMR type. The manuscript now includes the following: *“From our high-confidence callset of 5,297 SVs, we identified 543 nonredundant TEMRs (10.25%) across all three individuals (Fig. 1a). We identified an average of 263 TEMRs per sample (236 from PUR, 236 from CHS, and 316 from YRI) and they collectively affected an average of 795 kbp per sample. The 543 TEMRs consisted of 11 classes of TEs: Alu (397), LINE-1(96), ERVL-MaLR (14), ERV1 (11), ERVL (8), L2 (6), ERVK (3), MIR (2), SVA (2), TcMar-Mariner (1), TcMar-Tigger (1), hAT-Charlie (1) (Supplementary Table 1). Due to the prevalence of LINE-1 and Alu-mediated events, the difficulties in aligning ERVs and divergent transposons to consensus sequences, and the small number of TEMRs driven by non- Alu and LINE-1 categories precluding extensive mechanistic work, we focused on the two primary categories of TEMR in this study.”*

We also updated Fig. 1a with an additional category for “Other TEs” which comprises non- Alu or LINE-1 TEMRs.

We manually inspected the two SVA-driven TEMRs and found them to be HR driven:

CHR	POS	END	SVTYPE	SVLEN	HOMLEN	TE_5'	TE_3'	ORIENTATION
chr19	20148842	20152102	DEL	3260	256	SVA_F	SVA_D	SAME
chr4	150954410	150964467	DEL	10057	320	SVA_B	SVA_B	SAME

We also inspected the 48 Other TEMRs (those driven non-*Alu*, LINE-1 and SVA TEs) and found the median homology length at the breakpoint junction to be 4 bp and median size to be 795 bp. Due to the difficulties that preclude extensive mechanistic work with these classes of TEMRs, we added them to Supplemental Table 1, but did not further investigate the mechanisms or consequences of these events.

minor:

line 129: Provide the arguments the choice of three specific genomes.

We have added a sentence to reflect this comment, “We implemented our pipeline on Illumina and PacBio CLR data across three well-characterized genomes representative of: (1) population admixture, Puerto Rican HG00733 (PUR); (2) low diversity, Southern Han Chinese HG00514 (CHS); and (3) high diversity, Yoruban NA19240 (YRI)¹⁶.”

Additionally, these three individuals have been studied as a part of the 1000GP phase 3, HGSVC phase 1, and HGSVC phase 2 studies, which provides us with extensive genomic data ranging from short read sequencing to long read sequencing, Bionano genomics data and RNA-Seq that can be used for further understanding of TEMRs.

line 240: Is overrepresentation of younger TEs statistically significant?

Yes, we did find overrepresentation of younger TEs to be statistically significant, we have updated our results based on this query. The sentence reads: “Further, we found that *AluS* and *AluY* subfamilies were enriched within *Alu* TEMRs (*AluY*: 24.6% vs 11.8%, $p < 1e^{-22}$, two-tailed Fisher’s exact test and *AluS*: 66.8% vs 57.4%, $p < 1e^{-7}$, two-tailed Fisher’s exact test) and *L1PA* subfamilies were enriched within *LINE-1* TEMRs compared to GRCh38 (53.6% vs 12.6%, $p < 1e^{-41}$, two-tailed Fisher’s exact test) (Supplementary Table

5). This observation is in concordance with previous studies showing that younger TEs (fewer acquired mutations) are more likely to be involved in TEMRs¹⁹, and AluS TEMRs are enriched due to their relative abundance (678,131 AluS compared to 139,234 AluY elements in GRCh38³).”

We have also updated the Supplementary Table 5 to reflect the below information

TE family	subfamily	Count (this study)	Percentage (this study)	Count (GRCh38)	Percentage (GRCh38)	p-value
Alu	AluY	195	24.6	139234	11.8	<1e ⁻²²
	AluS	530	66.8	678131	57.4	<1e ⁻⁷
	AluJ	64	8.1	309536	26.2	
	Others	5	0.6	54171	4.6	
(397 Alu TEMRs)	Total	794		1181072		
LINE-1	L1HS	4	2.1	1620	0.2	<0.001
	L1PA	103	53.6	121006	12.6	<1e ⁻⁴¹
	L1M	60	31.3	747338	77.7	
	Others	25	13	92121	9.6	
	(96 LINE-1 TEMRs)	Total	192		962085	

Supplementary Table 5. Count of *Alu* and LINE-1 subfamilies involved in 493 TEMRs.

Figure 1: make sure you are using a colour blind palette.

We have updated the figure based on the Reviewers' comment and have checked that the hues are visible from a color-blind palette.

Figure 2: make the font for panels b. and c. the same size as in Figure 1 panel d.

We have corrected this inconsistency in figure labels.

Figure 4: consider using different points symbols for different genomes in panel d.

We have implemented the suggested change.

Reviewer #3 (Remarks to the Author):

This is a review of NCOMMS-22-16442-T, "Transposable element-mediated rearrangements are prevalent in human genomes." Herein, the authors use computational methods with deep sequencing data to identify structural variants (SVs) that have transposable elements (TEs) at the junctions in three human individuals, and then perform a series of analyses on these SVs, including some validation experiments with PCR/Sanger sequencing, and describing features of the SVs, such as homology length, degrees of sequence divergence, and sizes of the SVs. Among the 493 TE-associated SVs, most of them were deletions, but some were more complex, and an analysis of these complex events are shown, along with analysis of the propensity for these rearrangements in gene bodies. I found this analysis quite compelling, and its impact in part will be as a guide for mechanistic studies to determine the DNA repair pathways that mediate these events.

We thank the Reviewer for their positive assessment of the findings and methodologies used in our manuscript.

Major comments:

1. I suggest that the impact of the study could be enhanced by analyzing whether certain features of TE-mediated SVs show correlations, or not. Namely, perform correlation analysis with these parameters they describe: homology / non-homology and also homology length, vs. SV size, and vs. TE divergence. Does homology length, or % homology vs. non-homology correlate with SV size? With TE divergence? Does SV size correlate (positively or negatively) with TE divergence? Since the mechanism of repeat-mediated SVs can be affected by repeat distance and divergence (e.g. PMID: 32023454), such analysis would be very useful to the field. Of course, the lack of a correlation might be due to small sample size, but that caveat could be included in the text.

We thank the Review for this suggestion and have incorporated the correlation analysis among the features used to study TEMRs in this manuscript. We added the following paragraph to the result *"We grouped TEMRs based on their mechanism (HR / NHE), family (Alu / LINE-1) and orientation of the TE involved (Direct / Indirect) and performed correlation analysis among three main characteristics used to discern TEMR mechanisms: the length of the TEMR, the tract length of homology at the breakpoint junction, and the similarity between the two TEs involved in the rearrangement (Supplementary Table 7). The homology length and TE similarity among TEMR-HRs were the only categories with statistically significant correlation value for both Alu TEMRs (correlation = 0.19, $p < 0.001$, Spearman's correlation) and LINE-1 TEMRs (correlation = 0.49, $p < 0.01$, Spearman's correlation). We were unable to find any other statistically significant (positive/negative) correlations among the examined features and believe this could be due to the small sample size, most notably of LINE-1 TEMRs."*

The below table has been added to the supplementary material to summarize the correlation analysis (Supplementary Table 7). The highlighted rows indicate the features and categories that had a statistically significant result ($p < 0.01$).

Mechanism	TE	Orientation	Feature_1	Feature_2	r-value (spearman)	p-value (spearman)
HR	Alu	SAME	HOMLEN	PCT_SIMILARITY	0.1932	0.00026
			HOMLEN	SV_LENGTH	0.08868	0.09573
			PCT_SIMILARITY	SV_LENGTH	0.12577	0.01791
HR	LINE-1	SAME	HOMLEN	PCT_SIMILARITY	0.49028	0.0024
			HOMLEN	SV_LENGTH	-0.26838	0.1135
			PCT_SIMILARITY	SV_LENGTH	-0.24659	0.14711
NHE	Alu	SAME	HOMLEN	PCT_SIMILARITY	-0.34638	0.1143
			HOMLEN	SV_LENGTH	0.11569	0.60818
			PCT_SIMILARITY	SV_LENGTH	0.05251	0.81647
NHE	Alu	OPP	HOMLEN	PCT_SIMILARITY	0.23634	0.36111
			HOMLEN	SV_LENGTH	0.11883	0.64966
			PCT_SIMILARITY	SV_LENGTH	-0.2451	0.34305
NHE	LINE-1	SAME	HOMLEN	PCT_SIMILARITY	0.02663	0.88114
			HOMLEN	SV_LENGTH	0.13095	0.4604
			PCT_SIMILARITY	SV_LENGTH	0.29015	0.096
NHE	LINE-1	OPP	HOMLEN	PCT_SIMILARITY	0.02352	0.90922
			HOMLEN	SV_LENGTH	0.1013	0.62244
			PCT_SIMILARITY	SV_LENGTH	-0.32923	0.10052

Supplementary Table 7. Summary of the correlation analysis.

- For TE divergence, is it possible to relate this to overall degrees of TE divergence for different types of TEs in the genome? Namely, some context for the degrees of TE divergence could enhance the study.

We thank the Reviewer for this comment, we inspected the TE divergence across the reference genome and have updated the results accordingly. The section now reads as, “We inspected the percent divergence among Alu and LINE-1 elements across the reference genome using the RepeatMasker dataset from UCSC genome browser and compared that to the Alu elements and LINE-1 elements from our TEMR callset. We found that Alu and LINE-1 elements from our callset have a lower median divergence when compared to the Alu and LINE-1 elements present within the latest reference genome (Alu: 9.6% vs 11.9%; $p < 1e^{-7}$, Welch’s t-test, and LINE-1: 9.9% vs 21.6%; $p < 1e^{-7}$, Welch’s t-test).”

We have updated the initial section of the results and Supplemental Figure 1 based on the Reviewer’s comments. The section now reads,

“We implemented our pipeline on Illumina and PacBio CLR data across three well-characterized genomes representative of: (1) population admixture, Puerto Rican HG00733 (PUR); (2) low diversity, Southern Han Chinese HG00514 (CHS); and (3) high diversity, Yoruban NA19240 (YRI)¹⁶. Implementing a multi-caller approach with additional filters have enabled us to significantly reduce the number of false positive variants in our callset (Methods). Due to the repetitive nature of TEs and the technical difficulty it causes during variant calling, analyzing TEMRs without any stringent filtering could led to an unreliable analysis due to amount of false positive being discovered (Supplementary Fig. 1). We have demonstrated with our pipeline that an ensemble approach with simple filters can result in a reliable callset outside simple repeat regions, especially when using short-read HTS data (Supplementary Fig. 1). Additionally, we obtained phased HiFi assemblies for these three samples¹⁶, and used a new version of PAV¹⁶ for breakpoint homology. We merged the calls from all these methods and across all three individuals into a single nonredundant high-confidence callset of 5,297 SVs containing 4,997 deletions, 239 duplications and 61 inversions, with an average of 3,111 SVs per individual (Methods).”

REVIEWERS' COMMENTS

Reviewer #1 (Remarks to the Author):

The authors have done an excellent job of addressing the issues raised in my critique. I have no further concerns or comments.

Reviewer #2 (Remarks to the Author):

I am completely satisfied with the corrections and the authors' replies. I also appreciate the additional analyzes carried out at the request of the reviewers, which improved the quality of the manuscript. In my opinion, the manuscript represents a very high scientific level and should be accepted for publication in Nature Communications in its present form.

Reviewer #3 (Remarks to the Author):

This is a compelling and impactful study defining various features of transposable-element mediated rearrangements. The revisions have adequately responded to my concerns/suggestions.

One note - in Supplemental Table 7, I see that the authors set a cutoff of $P < 0.01$, but this eliminates the possible correlation between PCT_SIMILARITY and SV_LENGTH for ALU HR events ($P = 0.014$). This could be really interesting, because it might indicate that ALUs that are more similar are able to engage in HR events at larger distances. I can understand the reluctance of the authors to state this correlation due to the p-value, but I might recommend a 1-2 sentences that note this result, but with the caveats that the authors may include (like, "however this correlation did not meet a threshold of $p < 0.01$ " or whatever the concerns might be).

REVIEWERS' COMMENTS

Reviewer #1 (Remarks to the Author):

The authors have done an excellent job of addressing the issues raised in my critique. I have no further concerns or comments.

Reviewer #2 (Remarks to the Author):

I am completely satisfied with the corrections and the authors' replies. I also appreciate the additional analyzes carried out at the request of the reviewers, which improved the quality of the manuscript. In my opinion, the manuscript represents a very high scientific level and should be accepted for publication in Nature Communications in its present form.

We thank reviewers 1 and 2 for making our manuscript better.

Reviewer #3 (Remarks to the Author):

This is a compelling and impactful study defining various features of transposable-element mediated rearrangements. The revisions have adequately responded to my concerns/suggestions.

One note - in Supplemental Table 7, I see that the authors set a cutoff of $P < 0.01$, but this eliminates the possible correlation between PCT_SIMILARITY and SV_LENGTH for ALU HR events ($P = 0.014$). This could be really interesting, because it might indicate that ALUs that are more similar are able to engage in HR events at larger distances. I can understand the reluctance of the authors to state this correlation due to the p-value, but I might recommend a 1-2 sentences that note this result, but with the caveats that the authors may include (like, "however this correlation did not meet a threshold of $p < 0.01$ " or whatever the concerns might be).

We appreciate the reviewer's appreciation of our work, and have added a sentence to our results section "Characteristics of TEMR Breakpoints" based on the suggestion. The sentence reads as follows: *"We also found that the percent similarity between the two Alu elements involved in an HR event is positively correlated (correlation = 0.13, $p = 0.018$, Spearman's correlation) with the size of TEMRs, but this failed to meet our threshold for significance ($p < 0.01$)."*